# Imprinting fidelity in mouse iPSCs depends on sex of donor cell and medium formulation

Maria Arez[1,2,3], Melanie Eckersley-Maslin [4,5,6,7], Tajda Klobučar[3,8], João von Gilsa Lopes [3,9], Felix Krueger [10,11], Annalisa Mupo[4,11], Ana Cláudia Raposo[1,2,3], David Oxley[12], Samantha Mancino[1,2,3], Anne-Valerie Gendrel [3,13], Bruno Bernardes de Jesus[3,14] & Simão Teixeira da Rocha [1,2,3] ✉

Reprogramming of somatic cells into induced Pluripotent Stem Cells (iPSCs) is a major leap towards personalised approaches to disease modelling and cell-replacement therapies. However, we still lack the ability to fully control the epigenetic status of iPSCs, which is a major hurdle for their downstream applications. Epigenetic fidelity can be tracked by genomic imprinting, a phenomenon dependent on DNA methylation, which is frequently perturbed in iPSCs by yet unknown reasons. To try to understand the causes underlying these defects, we conducted a thorough imprinting analysis using IMPLICON, a high-throughput method measuring DNA methylation levels, in multiple female and male murine iPSC lines generated under different experimental conditions. Our results show that imprinting defects are remarkably common in iPSCs, but their nature depends on the sex of donor cells and their response to culture conditions. Imprints in female iPSCs resist the initial genome-wide DNA demethylation wave during reprogramming, but ultimately cells accumulate hypomethylation defects irrespective of culture medium formulations. In contrast, imprinting defects on male iPSCs depends on the experimental conditions and arise during reprogramming, being mitigated by the addition of vitamin C (VitC). Our findings are fundamental to further optimise reprogramming strategies and generate iPSCs with a stable epigenome.

The seminal studies from Takahashi and Yamanaka over a decade ago demonstrated the possibility to revert a somatic cell to a stem-like state through the induction of a defined set of transcription factors[1]. This opened the prospect to reprogram patient-derived cells, helping to elucidate novel pathological mechanisms underlying human diseases and to reveal new therapeutic drugs. Furthermore, clinical trials involving iPSCs in cell-replacement therapies have already been initiated[2,3].

Despite being a revolutionary technology, several challenges persist that limit the current range of iPSC applications. First, iPSC reprogramming remains a highly inefficient process, with most of the cells either failing or achieving only partial reprogramming[4–7]. Second,

correctly reprogrammed iPSCs exhibit very heterogeneous responses to specific differentiation cues, which compromise their use in disease modelling and clinical applications[8–10]. The lack of consistency between different iPSCs can be attributed to the natural genetic variability among individual donors[11], as well as recurrent genetic aberrations in iPSCs[12–14]. However, the persistence of such heterogeneous behaviour in isogenic iPSC lines also suggests a role of epigenetic variability in this phenomenon[10,15–17].

Epigenetic disparity, mostly studied at the DNA methylation level, can originate from: (1) failure to fully reset the somatic memory of the donor cell[8,9,18–20]; (2) acquisition of aberrations during the reprogramming process[16–19,21–24]; (3) epigenetic adaptation to long-term in vitro

culturing[25,26]. The latter two likely affect processes that rely on strict maintenance of DNA methylation, such as genomic imprinting[24,27–29], which serves as a good readout to study epigenetic fidelity in iPSCs.

Genomic imprinting is a parent-of-origin-specific epigenetic mechanism that controls the monoallelic expression of ~150 genes in the mammalian genome. Unlike the majority of genes, imprinted genes are biased or exclusively expressed from the maternal or paternal chromosomes. These genes are important regulators in prenatal growth and development, as well as, in postnatal brain functions and metabolic pathways[30]. The dysregulation of imprinted genes has been associated with several developmental and behavioural disorders, such as Beckwith-Weidemann, Angelman and Prader-Willi syndromes[31–33]. Most imprinted genes are located in close proximity in defined genomic loci, called imprinted clusters. Mammalian genomes present around 25 imprinted clusters, that contain a *cis*-acting regulatory element, known as imprinting control region (ICR), which co-regulates imprinting expression of multiple genes. This region is dense in CpG dinucleotides and displays opposite DNA methylation patterns between the paternally and maternally inherited alleles. This methylation pattern is set up during gametogenesis and stably maintained in somatic cells, with 22 of the ICRs being methylated on the maternal allele and 3 on the paternal allele in the mouse genome[30]. Disturbances in the parental allele-specific methylation at ICRs perturb monoallelic expression and are thus one of the main causes of imprinting disorders[34]. Therefore, it is of utmost importance that proper parental-specific methylation at ICRs is maintained in iPSCs.

Previous studies have revealed imprinting defects in both mouse and human iPSCs[16,17,24,27–29,35]. Importantly, these errors persist and are never rescued upon differentiation[17,35], which is troublesome for iPSC applications in translational and clinical research. The first reported imprinting defect was the aberrant hypermethylation of the maternal allele in the *Dlk1-Dio3* cluster leading to silencing of maternally expressed non-coding RNAs in mouse iPSCs[24]. This was shown to compromise their pluripotent properties[24,36,37]. Interestingly, defects in *Dlk1-Dio3* locus could be corrected by using ascorbic acid, also known as vitamin C (VitC)[38], presumably due to an increased demethylating activity of the TET dioxygenase enzymes[39]. Later studies showed that the repertoire of imprinting defects extends to other imprinted loci in mouse iPSCs[27–29]. Interestingly, the extent and nature (hypermethylation versus hypomethylation) of these defects varied significantly between studies (Supplementary Table 1). Whether these contrasting results are caused by differences in the sex of donor cells and/or the reprogramming protocols used, or by limitations of the systems chosen for imprinting analyses is unknown (Supplementary Table 1). Several shortcomings can be attributed to those studies which prevent direct comparison: (1) unavailability of single nucleotide polymorphisms (SNPs) to distinguish the two parental alleles for imprinting analysis; (2) low number of iPSC lines studied; (3) lack of information or use of iPSCs of only one sex; (4) use of a single culture condition for reprogramming; (5) limited number of imprinted regions assessed.

Here, we present a thorough analysis of imprinting defects in isogenic murine female and male hybrid iPSCs in different culture conditions using the ultra-deep IMPLICON approach to screen for imprinting methylation. Our results show that iPSCs harbour multiple imprinting errors that are dictated by sex of the donor cell and medium conditions. As imprinting defects are maintained in differentiated derivatives of iPSCs, our findings are fundamental for continuous improvement of reprogramming protocols aiming at generating epigenetically faithful iPSCs.

## Results

### Generation of F1 hybrid iPSCs in serum-free conditions

To systematically address the imprinting status in mouse iPSCs, we generated male and female isogenic hybrid iPSC lines from mouse embryonic fibroblasts (MEFs) derived from a cross between a female "*reprogrammable*" i4F mouse carrying a doxycycline-inducible polycistronic Yamanaka cassette (on a C57BL/6J background; named i4F-BL6 herein)[40,41] and a male from the *Mus musculus castaneus* (CAST/EiJ - named CAST herein) strain. The use of F1 hybrid cells from genetically distant mouse strains allows the distinction of parental alleles based on SNPs present at both ICRs and genes[42,43]. Unfortunately, we were unable to obtain MEFs from the reciprocal cross within the time-frame of this work. For this reason, we focused our analysis on loci for which imprinting has previously been shown not to be perturbed by the direction of the cross either in tissues or in mouse iPSCs[27,43,44].

Both female and male MEFs were first reprogrammed in serum-free conditions using a medium supplemented with Knockout Serum Replacement (KSR) in the presence of doxycycline (DOX) for 12 days (Fig. 1A) as previously described[41]. After 12 days of DOX induction, we picked 6 independent colonies with the typical dome-shaped morphology from each sex. The cells were maintained in the same medium without DOX until around day 50 to ensure the generation of totally reprogrammed iPSC lines (Fig. 1A). All analyses were performed for all iPSC lines at this stage and in the subsequent two-to-three passages.

To validate the newly KSR-derived iPSC lines (KSR-iPSCs), we screened for the expression of three pluripotent markers (*Pou5f1*, *Nanog* and *Esrrb*). All KSR-iPSCs, but not the parental MEFs, expressed the markers at similar levels to mouse embryonic stem cells (ESCs) cultured in serum (JM8.F6 male ESCs)[45] or 2i (MEK and GSK3 inhibition) conditions (TX1072 female ESCs from a BL6 female x CAST male cross – TX 2i ESCs)[46] (Supplementary Fig. 1A). We strengthened our transcriptional characterization by performing RNA sequencing (RNA-seq) in three biological replicates for two female (F KSR2 and F KSR4) and two male (M KSR3 and M KSR5) iPSC lines, TX 2i ESCs, as well as female MEFs. Clustering analysis based on expression profile demarcates differences between cell types (MEFs versus iPSCs/ESCs) and culture conditions (KSR versus 2i) (Fig. 1B). Analysis of a selected set of pluripotency genes showed consistent expression in iPSCs and TX 2i ESCs, but not in female MEFs (Fig. 1C). This was confirmed at the protein level by immunofluorescence (IF) for SSEA-1, OCT4 and NANOG (Supplementary Fig. 1B). We also checked that our female and male KSR-iPSCs exhibit a normal karyotype (Supplementary Fig. 1C). Consistent with this, RNA-seq counts and differential gene expression analysis of females compared to male iPSCs indicate the presence of two active X chromosomes in our female iPSCs (Supplementary Fig. 1D, E; Supplementary Data 1). When female and male KSR-iPSC lines were subcutaneously injected into immune-deficient NOD SCID γ (NSG) mice[47], they all formed teratomas composed by cells belonging to the three germ layers (endoderm, mesoderm and ectoderm) and the occasional presence of trophectoderm-derived trophoblast giant cells, confirming their full differentiation potential (Fig. 1D). All together, these results show that we successfully generated multiple pluripotent and karyotypically normal female and male hybrid KSR-iPSCs.

### KSR-iPSCs show loss of methylation at imprinted regions

To screen for imprinting methylation fidelity in KSR-iPSCs, we employed allele-specific IMPLICON previously validated on F1 tissue samples from reciprocal BL6 x CAST crosses[44]. This method combines bisulfite treatment of genomic DNA with amplicon high-throughput sequencing with a de-duplication step to generate base-resolution datasets with high coverage and allelic discrimination of the original DNA molecules. We focused on 13 imprinted clusters together with 2 unmethylated and 1 methylated control regions and looked in KSR-iPSCs, the original MEFs and in the TX 2i ESC line expected to have erased methylation imprints due 2i-induced demethylation[48–51]. Reassuringly, no differences between the two parental alleles were observed for both unmethylated (*Sox2* and *Klf4* genes) and methylated controls (*Prickle1* gene) in female and male MEFs and respective KSR-iPSCs, as well as, for the TX 2i ESCs (Supplementary Fig. 2A;

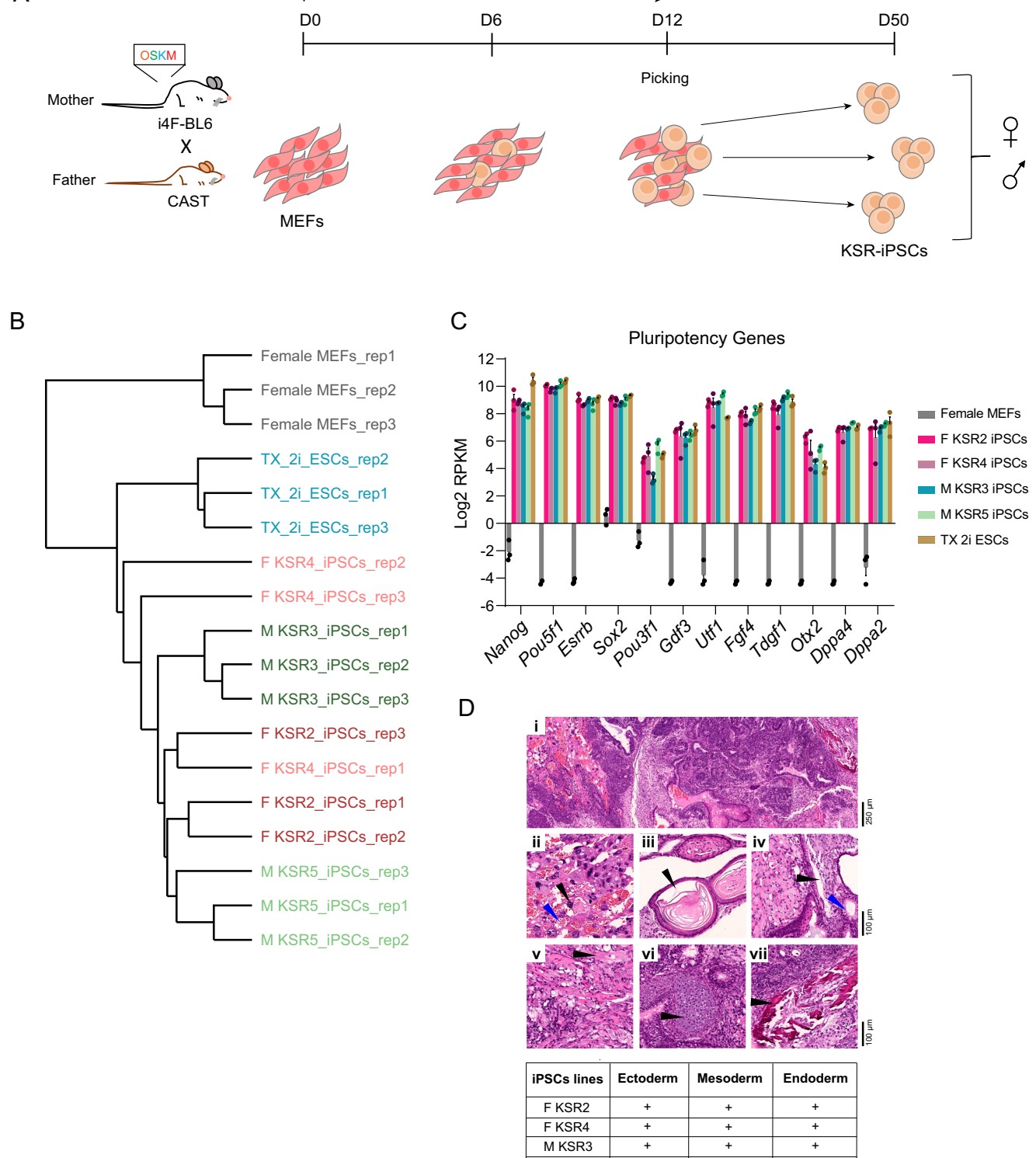

Supplementary Data 2). For *Prickle1*, a slight drop in DNA methylation levels was observed for female KSR-iPSCs (-50-70%) in both alleles compared to parental MEFs (>90%), while it was completely lost in the TX 2i ESC line (<10%), likely to be caused by 2i-induced demethylation[48–51].

For the 13 imprinted regions analysed, at both female and male MEFs we observed the expected allele-specific methylation pattern at ICRs, with 11 of the regions displaying methylation on the maternal allele (PWS/AS, *Peg3, Gnas, Commd1-Zrsr1, Mcts2-H13, Kcqn1-Kcnq1ot1,*

*Mest/Peg1, Plagl1/Zac1, Grb10, Igf2r* and *Impact*) and two regions on the paternal allele (*Igf2-H19* and *Dlk1-Dio3*) (Fig. 2A; Supplementary Fig. 2B; Supplementary Data 2). This allele-specific pattern was mostly erased in the TX 2i ESCs, confirming that the 2i-induced demethylation does not spare imprinted loci as previously reported[52,53]. Strikingly, the expected imprinting methylation status (>75 % on one and <25 % on the other allele) was not preserved in either female or male KSR-iPSCs. First, we observed that reduction in methylation from the methylated ICRs was a common trend to all KSR-iPSCs (Fig. 2A; Supplementary

**Fig. 1 | Generation of F1 hybrid KSR-iPSCs. A** Schematic representation of the reprogramming protocol; briefly, a transgenic "*reprogrammable*" female mouse on a C57BL/6J genetic background (i4F-BL6) was crossed with a *Mus musculus castaneus* (CAST) male mouse to generate E13.5 F1 hybrid embryos from which mouse embryonic fibroblasts (MEFs) were obtained. MEFs were reprogrammed by induction of the polycistronic Yamanaka cassette (*Oct4/Sox2/Klf4/c-Myc* - OSKM) in the presence of doxycycline (DOX) for 12 days. Individual clones of mouse induced pluripotent stem cells reprogrammed in Knockout Serum Replacement medium (KSR-iPSCs) were picked on day 12 and expanded until approximately day 50. **B** Clustering analysis of the normalised RNAseq counts for all the biological triplicates of female MEFs, female (F KSR2 and F KSR4) and male (M KSR3 and M KSR5) iPSCs and TX 2i ESCs. **C** Expression analysis by RNAseq of a panel of pluripotent genes in female MEFs, female (F KSR2, F KSR4), male (M KSR3 and M KSR5) iPSCs and TX 2i ESCs. The graph shows the average Log2 Reads Per Kilobase per Million mapped reads (RPKM) expression values ± Standard Deviation (SD) from biological

triplicates of each sample. Source data are provided as a Source Data file. **D** Table and representative H&E staining of teratomas after subcutaneous injection of $2 \times 10^6$ cells into the flanks of NSG mice. iPSCs efficiently contribute to ectoderm, mesoderm, endoderm and occasionally trophectoderm. **i** Low magnification of a mature teratoma, scale bar represents 250 μm. **ii** Trophectoderm-derived trophoblast giant cells (black arrowhead), associated with large vascular spaces (blue arrowhead), characteristic of placental tissue. **iii** Ectodermal components corresponding to squamous epithelium (black arrowhead). **iv** Endodermal components corresponding to respiratory-type epithelium, including ciliated (black arrowhead), and mucin-producing goblet cells (blue arrowhead). **v, vi, vii**, Mesodermal components (black arrowhead) corresponding to muscle, cartilage, and bone, respectively; ii-vii scale bar represents 100 μm. Table summarises the successful generation of teratomas with tissues from the three germ layers from F KSR2, F KSR4, M KSR3 and M KSR5 iPSCs. Two teratomas per cell line were generated and analysed by H&E staining.

Fig. 2B; Supplementary Data 2), and affected both maternally and paternally methylated ICRs. Second, these hypomethylation defects were more pronounced in female versus male KSR-iPSCs. This is particularly noticeable looking at averaged methylation levels of female and male KSR-iPSCs (Fig. 2B), with 5 loci (*Gnas*, PWS/AS, *Igf2-H19*, *Dlk1-Dio3* and *Commd1-Zrsr1*) consistently more affected in female KSR-iPSCs (Fig. 2B). Third, we also noticed some iPSC-to-iPSC variation in methylation levels at imprinted regions (Fig. 2A, B; Supplementary Fig. 2B; Supplementary Data 2) matching previous observations[27–29].

Hypomethylation defects ranged from a milder reduction in methylation (30-70%) to the complete loss of methylation (< 10%) at the methylated allele (Fig. 2A; Supplementary Fig. 2B). Milder cases could be explained either by partial methylation loss at ICRs in all cells equally or by cellular heterogeneity within the same iPSC line, having some cells completely lost DNA methylation at ICRs, while others maintained it intact. Thanks to the single-molecule resolution of IMPLICON, we could see that the majority of these mild cases can indeed be explained by cellular heterogeneity. At the *Gnas* and *Kcnq1-Kcnq1ot1* loci, a partition between unmethylated and methylated reads could be seen for the normally methylated maternal ICRs (Supplementary Fig. 2C), suggesting that some cells have lost DNA methylation at these elements, while others have retained it. Overall, our results show that both female and male KSR-iPSCs have substantial hypomethylation defects suggestive of a general tendency of imprinting erasure under these conditions.

### Hypomethylation affects imprinted expression in KSR-iPSCs

We next examined whether ICR hypomethylation disrupts the normal allele-specific expression of imprinted genes by analysing our RNA-seq datasets taking advantage of the strain-specific SNPs. From a list of known murine imprinted genes[30] with stable imprinting in BL6 x CAST reciprocal crosses, we shortlisted 17 imprinted genes (13 paternally and 4 maternally expressed) transcribed from a single allele in MEFs (ratio: > 90%:10%), expressed in all iPSC replicates (Log2 RPKM > 1) and with sufficient allelic resolution (normalised cumulative SNP-specific read counts > 5) in at least two of the three replicates (Fig. 3A; Supplementary Data 3). Consistent with the widespread hypomethylation defects in KSR-iPSCs and TX 2i ESCs (Fig. 2A, B; Supplementary Fig. 2B), we no longer found the expected parental allele-specific expression for the vast majority of imprinted genes (Fig. 3A). Most imprinted genes became biallelically expressed with three genes (e.g., *Meg3, Peg3* and *Plagl1*) even showing bias towards expressing the originally silenced allele. In the F KSR2 iPSC line, the apparent switch of parental allele expression in the *Sgce, Peg10* and *Mest* genes (Fig. 3A) was later confirmed to be caused by maternal uniparental disomy of chromosome 6 by allelic-specific RNA-seq counts (Supplementary Fig. 3A-C).

The loss of monoallelic expression can be exemplified for the maternally *H19* expressed gene in female and male KSR-iPSCs, as well as in TX 2i ESCs (Fig. 3B). We also found that loss of monoallelic

expression was dependent on the degree of the hypomethylation defect. For example, in the PWS/AS cluster, the paternally expressed *Snrpn* gene becomes biallelic in F KSR2 iPSCs and TX 2i ESCs due to loss of maternal ICR methylation, but remains monoallelically expressed in the non-affected M KSR5 iPSCs (Fig. 2A and 3B). Alterations in allele-specific expression were broadly consistent with the methylation changes at ICRs (Fig. 2A, B; Fig. 3A, B; Supplementary Data 2), revealing that both female and male KSR-iPSCs are incapable of maintaining their imprinted expression.

### Imprinting defects in FBS-iPSCs are sex-specific

Given our findings of major hypomethylation defects at ICRs in KSR-iPSCs, we next asked whether switching to different culture conditions would change this phenotype. We applied the same reprogramming paradigm but using the classical ESC medium conditions based on Foetal Bovine Serum (FBS) (Fig. 4A). We successfully generated 5 female and 5 male FBS-derived iPSCs (FBS-iPSCs) that exhibited normal ESC-like morphology, expressed pluripotent stem cell markers, were able to differentiate into the three germ layers and presented a normal karyotype (Supplementary Fig. 4A–C). An exception to that was the F FBS1 line that lost one chromosome. This is likely to be the X chromosome, commonly absent in female murine ESCs/iPSCs[54,55], as X-linked gene expression in F FBS1 iPSCs did not differ from male iPSCs and was half that found in other female KSR- and FBS-iPSCs (Supplementary Fig. 4D).

Allele-specific IMPLICON was then performed for the same 13 imprinted regions and methylated/unmethylated controls. No allelic differences were observed for both the unmethylated (*Sox2* and *Klf4*) and methylated (*Prickle1*) control regions (Supplementary Data 2). Female FBS-iPSCs exhibited lower levels of methylation in both alleles for *Prickle1* as previously seen in female KSR-iPSCs (Supplementary Fig. 2A; Supplementary Data 2). Strikingly, we found different imprinting outcomes in female versus male FBS-iPSCs. Similar to KSR-iPSCs (Fig. 2A, B), female FBS-iPSCs showed a strong tendency to demethylate the methylated ICRs (Fig. 4B; Supplementary Fig. 4E, F; Supplementary Data 2), resulting in biallelic expression of imprinted genes, as illustrated for the *H19* and *Snrpn* genes (Fig. 4C). The XO F FBS1, as well as F FBS4, did not escape to a generalised hypomethylated phenotype, but a milder effect was seen in a few loci (Fig. 4B; Supplementary Data 2). The unmethylated ICRs in female iPSCs were untouched, but a mild increase at the *Dlk1-Dio3* cluster was consistently seen in a few female FBS-iPSCs (Fig. 4B; Supplementary Fig. 4E; Supplementary Fig. 5A; Supplementary Data 2). However, this does not seem to improve the abnormal biallelic expression of *Meg3* gene seen for female KSR- and FBS-iPSCs (Supplementary Fig. 5A). In contrast to female, male FBS-iPSCs preserved the typical imprinting methylation status for 11 of the 13 imprinted regions (respectively, > ~70% and < ~15% of methylation levels at the methylated and unmethylated ICRs) and showing correct monoallelic expression of imprinted genes

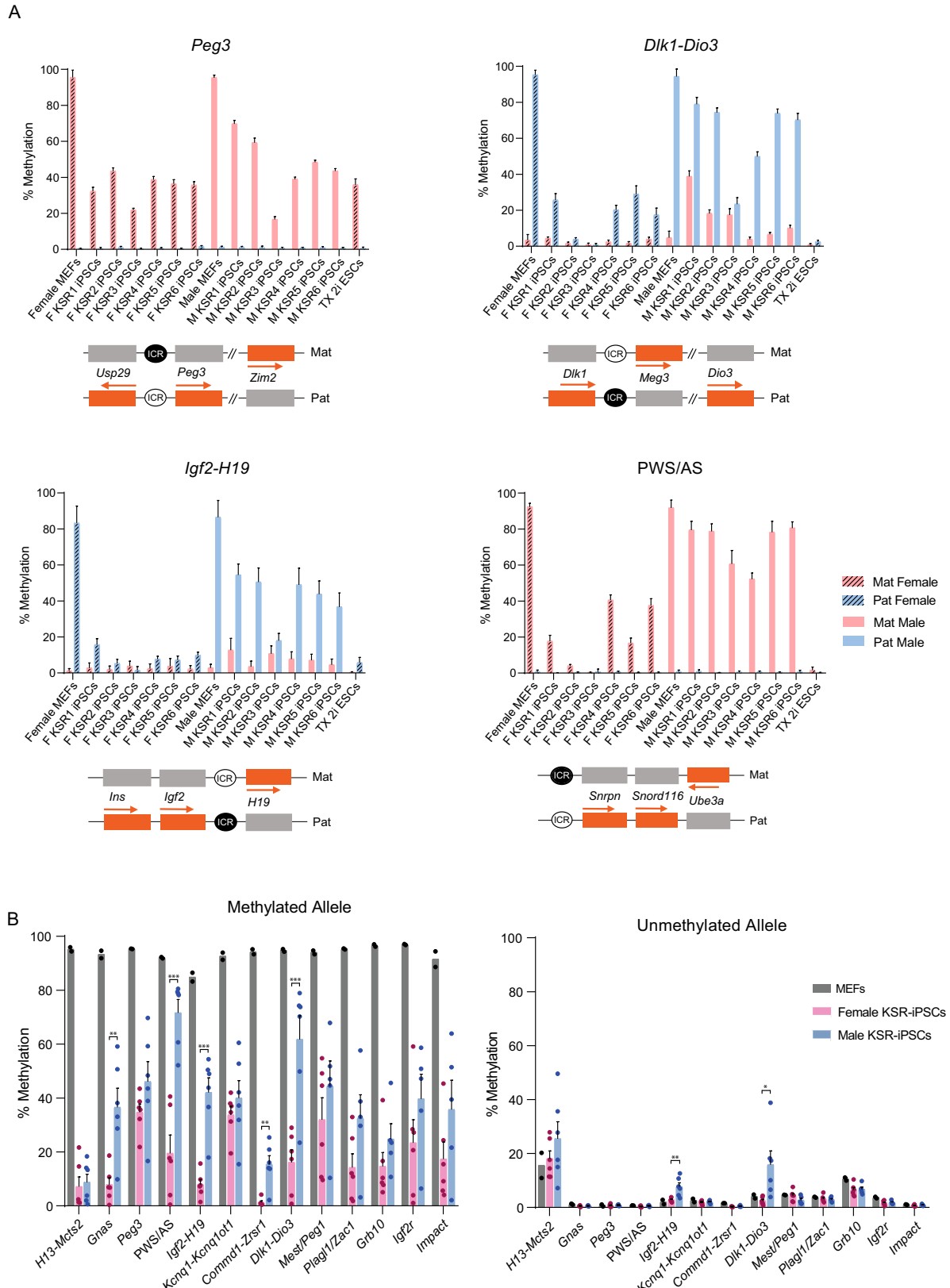

(Supplementary Fig. 4E; Fig. 4C). Only the two paternally methylated regions (*Dlk1-Dio3* and *Igf2-H19*) analysed exhibit signs of hypermethylation of the unmethylated ICR (Fig. 4B; Supplementary Fig. 4E). The *Igf2-H19* cluster shows a tendency to gain methylation at the unmethylated allele (Fig. 4B; Supplementary Fig. 4E, F), however, this does not seem to perturb the monoallelic expression of *H19* (Fig. 4C). A stronger gain of methylation was seen in the *Dlk1-Dio3* region, where 4 out of 5 iPSC clones showed equivalent high methylation levels in both parental alleles (Fig. 4B). While M FBS1 iPSCs, with normal methylation at the *Dlk1-Dio3* ICR showed monoallelic *Meg3* expression, the hypermethylated M FBS5 line showed lack of *Meg3* expression (Supplementary Fig. 5A). Considering both female and male KSR- and FBS-iPSCs, we

**Fig. 2 | Hypomethylation defects in KSR-iPSCs. A** Methylation analysis of *Peg3*, *Dlk1-Dio3*, *Igf2-H19* and PWS/AS ICRs in female and male MEFs, female (F KSR1-6) and male (M KSR1-6) iPSCs and TX 2i ESCs; Each graph represents the mean percentage ± SD methylation levels measured at each CpG within different genomic regions per parental allele for each sample (number of CpG per locus - *Peg3*: *n* = 24; *Dlk1-Dio3*: *n* = 27; *Igf2-H19*: *n* = 16; PWS/AS: *n* = 15); Scheme on the bottom of each graph represents the normal methylation status of each ICR in the correspondent regions (white circle – unmethylated ICR; black circle – methylated ICR; Mat – maternal allele; Pat – paternal allele; orange rectangles – expressed genes; grey

rectangles – silenced genes; regions are not drawn to scale). Source data are provided as Supplementary Data 2. **B** Average percentage of methylation at methylated and unmethylated alleles of ICRs in parental MEFs (*n* = 2 biological independent cell lines), female and male KSR-iPSCs (*n* = 6 each biological independent cell lines); Graph represents the mean ± SEM methylation levels measured at each CpG within different genomic regions per parental allele for each group of samples. Statistically significant differences between female and male KSR-iPSCs are indicated as * *p* < 0.05; ** *p* < 0.01; *** *p* < 0.001 (unpaired two-tailed Student's *t*-test). Source data are provided as Supplementary Data 2.

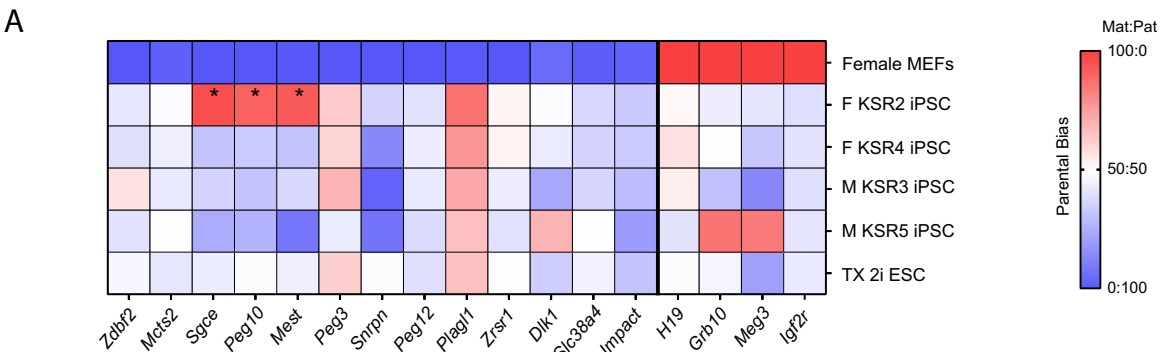

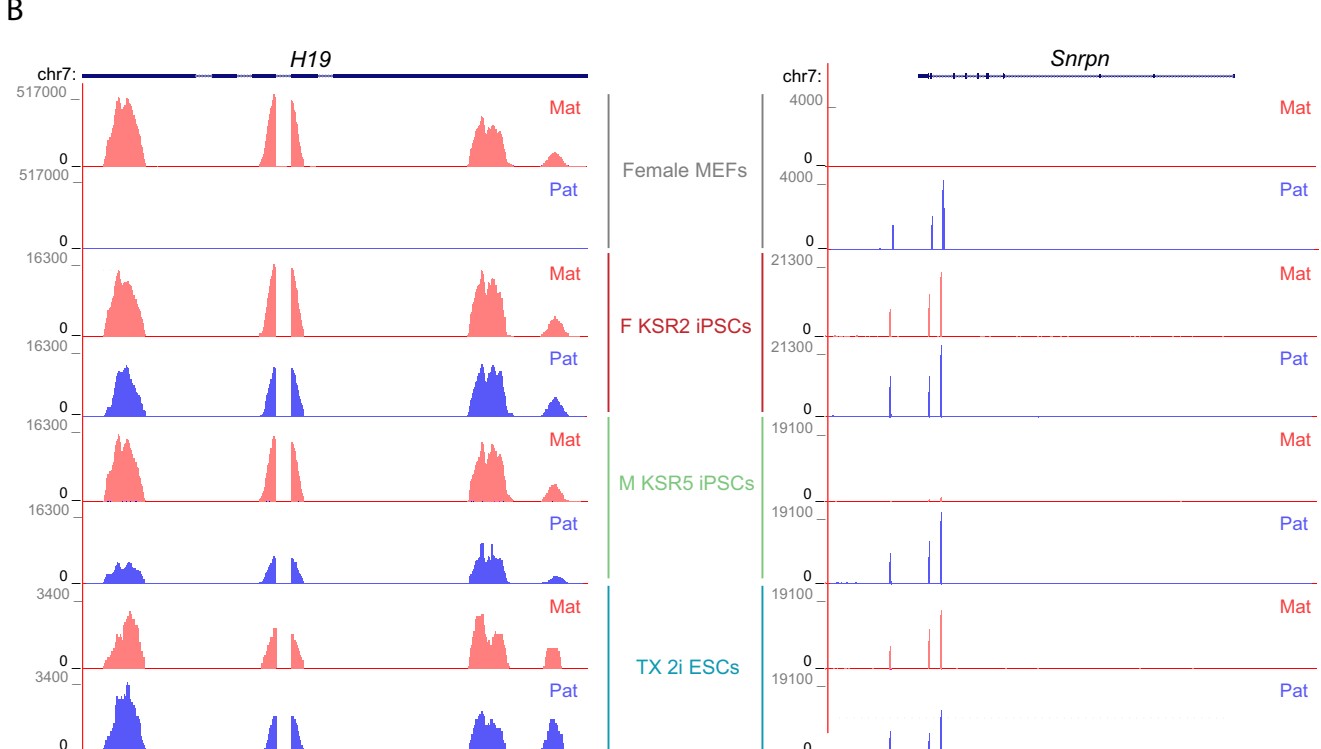

**Fig. 3 | Dysregulation of imprinted expression in KSR-iPSCs. A** Heatmap representing the mean parental allele-specific expression biases of imprinted genes in the biological triplicates of female MEFs, female (F KSR2 and F KSR4) and male (M KSR3 and M KSR5) iPSCs and TX 2i ESCs. Asterisks signal genes exhibiting parental-specific expression biases induced by maternal uniparental disomy of chromosome 6 on F KSR2 iPSCs (see Supplementary Fig. 3A). Source data are provided as

Supplementary Data 3. **B** Genome browser plots showing allele-specific reads in the *H19* and *Snrpn* genes in female MEFs, F KSR2 and M KSR5 iPSCs and TX 2i ESCs. Maternal (Mat) reads are shown in red and paternal (Pat) reads in blue. Note that due to SNP distribution not all parts of the transcript can be assessed in an allele-specific manner.

observed that the *Dlk1-Dio3* cluster was the most labile locus among the 13 imprinted loci investigated in this study (Fig. 5A) with diverse methylation outcomes affecting the allelic preference and levels of the *Meg3* gene located downstream of the *Dlk1-Dio3* ICR (Supplementary

Fig. 5A,B; Supplementary Data 2). We also found that *Meg3* levels were consistently higher in KSR versus FBS medium, but correlated negatively with the DNA methylation level at *Dlk1-Dio3* ICR (Supplementary Fig. 5B). These results are important given the role attributed to this locus for the

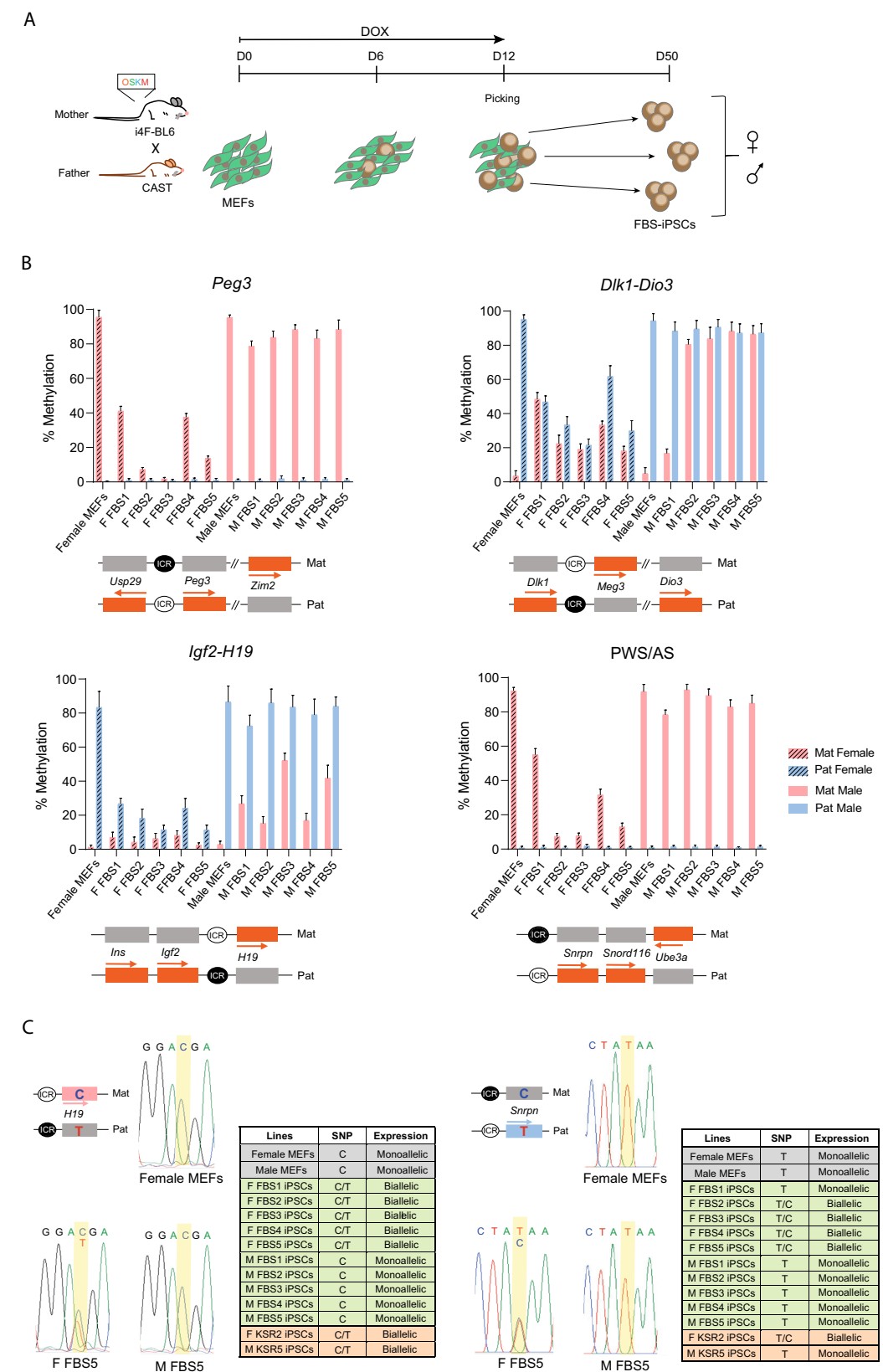

developmental potential of both mouse and human pluripotent stem cells[24,37,56].

In summary, our results revealed marked differences in imprinting fidelity between female and male iPSCs reprogrammed under standard serum conditions. We could detect sex-specific differences in the nature of imprinting defects with hypomethylation linked to female, and hypermethylation defects at specific loci linked to male FBS-iPSCs.

## Sex and medium formulation impact on global 5mC/5hmC levels

To understand the causes of these different imprinting abnormalities found by IMPLICON (Fig. 5A), we then evaluated whether global DNA

**Fig. 4 | Imprinting methylation defects in FBS-iPSCs. A** Schematic representation of the reprogramming protocol; briefly, a transgenic "*reprogrammable*" female mouse on a C57BL/6J genetic background (i4F-BL6) was crossed with a *Mus musculus castaneus* (CAST) male mouse to generate E13.5 F1 hybrid embryos from which mouse embryonic fibroblasts (MEFs) were collected. MEFs were reprogrammed by induction of the polycistronic Yamanaka cassette (*Oct4/Sox2/Klf4/c-Myc* - OSKM) in the presence of doxycycline (DOX) for 12 days. Individual mouse induced pluripotent stem cells reprogrammed in Foetal Bovine Serum medium (FBS-iPSCs) clones were picked at day 12 and expanded until approximately day 50. **B** Methylation analysis of *Peg3, Dlk1-Dio3, Igf2-H19* and PWS/AS ICRs in male and female MEFs (note: same data as in Fig. 2A for MEFs), female (F FBS1-5) and male (M FBS1-5) FBS-iPSCs; Each graph represents the mean percentage ± SD methylation levels measured at each CpG within different genomic regions per parental allele for each sample (number of CpG per locus - *Peg3*: *n* = 24; *Dlk1-Dio3*: *n* = 27; *Igf2-H19*: *n* = 16; PWS/AS: *n* = 15); Scheme on the bottom of each graph represents the normal methylation status of each ICR in the correspondent regions (white circle – unmethylated ICR; black circle – methylated ICR; Mat – maternal allele; Pat – paternal allele; orange rectangles – expressed genes; grey rectangles – silenced genes; regions are not drawn to scale. Source data are provided as Supplementary Data 2. **C** Allelic expression of *H19* and *Snrpn* genes assayed by RT-PCR followed by Sanger sequencing. Chromatograms are shown for female MEFs, F FBS5 and M FBS5 iPSCs; Table summarises the allele-specific expression for all the FBS-iPSCs as well as female and male MEFs, F KSR2 and M KSR3 iPSCs. Schemes on the left of the female MEFs chromatograms represent the normal imprinting profile of both *H19* and *Snrpn* and the associated single nucleotide polymorphism (SNP) for each allele (white circle – unmethylated ICR; black circle – methylated ICR; Mat – maternal allele; Pat – paternal allele; pink rectangle – maternally *H19* expressed gene; blue rectangle – paternally *Snrpn* expressed gene; grey rectangles – silenced genes; regions are not drawn to scale).

methylation differs between female and male KSR- and FBS-iPSCs. First, we monitored the global 5-methylcytosine (5mC) levels by Liquid Chromatography with tandem mass spectrometry (LC-MS/MS) in female and male KSR- and FBS-iPSCs and TX 2i ESCs. LC-MS measurements revealed that female iPSCs showed a marginal tendency for lower 5mC levels when compared to male iPSCs irrespective of the medium conditions (Fig. 5B). These are consistent with previous findings showing that female mouse ESCs/iPSCs have lower global 5mC levels[54,55,57]. On the other hand, the modest shift towards increased 5mC levels from KSR to FBS conditions for both female and male iPSCs was not found to be statistically significant (Fig. 5B). As expected, both female and male KSR- and FBS-iPSCs have considerably higher 5mC levels than the female TX 2i ESCs (Fig. 5B), consistent with minimal methylation levels associated with 2i medium conditions[50,51]. Therefore, our iPSCs showed relatively mild changes in overall DNA methylation and yet have marked imprinting defects.

To complement this analysis, we investigated the methylation levels at abundant retrotransposon elements in the genome, known to be highly methylated in ESCs in serum conditions[58]. We studied the intracisternal A particle (IAP), the most abundant Long Terminal Repeat (LTR) from the class II of endogenous retroviruses (ERVs)[59] as well as, two subfamilies of young Long Interspersed Nuclear Element 1 (LINE1), LINE1-A (L1-A) and LINE1-T (L1-T), which comprise the most frequent class of non-LTR elements in the mouse genome[60]. By performing quantitative bisulfite-pyrosequencing at these retrotransposon elements[58], we showed that each class of retroelements behaved differently according to the sex or medium formulation (Fig. 5C). IAPs suffered a decrease of DNA methylation in female iPSCs, which was potentiated, surprisingly, by FBS medium. L1-A methylation levels were also decreased in female KSR and FBS-iPSCs (Fig. 5C). These fluctuations in DNA methylation were modest compared to the severe demethylation seen for the TX 2i ESCs (Fig. 5C) as expected[58]. Interestingly, L1-T levels clearly dropped in both female and male KSR-iPSCs, reaching similarly low methylation levels to TX 2i ESCs (Fig. 5C). In summary, while a modest decrease in the level of 5mC and in methylation of IAPs and L1-A elements could be discerned for female iPSCs (Fig. 5B, C), a strong effect of KSR medium in demethylating L1-T retrotransposons was detected (Fig. 5C). In conclusion, with the exception of L1-T retrotransposons, ICRs are among the most affected regions upon reprogramming with more pronounced effects on DNA methylation than the ones observed at repetitive elements.

To understand the reasons behind global changes in DNA methylation and at imprinted regions in female versus male KSR- and FBS-iPSCs, we analysed expression levels of genes involved in DNA methylation and their co-factors (*Dnmt3a, Dnmt3b, Dnmt3l, Dnmt1* and *Uhrf1*), demethylation (*Tet1, Tet2* and *Tet3*) and imprinting protection (*Zfp57, Trim28* and *Dppa3*). No differences were detected between female and male iPSCs, suggesting that methylation variations at ICRs and globally between sexes are not caused by expression changes in

the core DNA methylation/demethylation machinery or imprinting protecting factors (Supplementary Fig. 6). In contrast, medium formulations had an impact on the expression of these genes: increased levels of *Tet1, Tet2* and *Dnmt3a* in KSR medium and *Dnmt3l* and *Uhrf1* in FBS medium (Supplementary Fig. 6). The increase of *Dnmt3a* levels in KSR medium could be offset by decreased levels of its co-factor *Dnmt3L*[61], which could explain why KSR medium does not cause increased DNA methylation. On the other hand, increased levels of the *Tet1* and *Tet2* dioxygenase enzymes in KSR medium could impact by converting 5mC to 5-hydroxymethylcytosine (5hmC). Interestingly, KSR-iPSCs had higher 5hmC levels than FBS-iPSCs, an effect especially evident for male KSR-iPSCs (Fig. 5D). This effect becomes clearer for both female and male iPSCs when the 5hmC/5mC ratio is taken into consideration (Fig. 5E). This effect could be due to the increased levels of *Tet1* and *Tet2*, but also to the presence of VitC in the KSR formulation which is known to increase the conversion of 5mC to 5hmC by TET dioxygenase enzymes[39]. In conclusion, our analysis suggests that imprinting defects in female iPSCs seem not to depend on changes in the basic DNA methylation or imprinting protection pathways. In contrast, imprinting methylation in male iPSCs might be responsive to medium conditions through fluctuations in the expression and activity of TET dioxygenase enzymes according to the medium formulation.

## Imprinting errors in male FBS-iPSCs arise during reprogramming

Female iPSCs consistently exhibit hypomethylation defects irrespective of the medium formulation. In contrast, male iPSCs have distinct imprinting defects when reprogrammed and maintained in KSR versus FBS medium (Fig. 5A). To understand whether medium composition by itself has an impact on imprinting in male iPSCs, we decided to culture FBS-iPSCs (M FBS1 and M FBS5) in KSR conditions and KSR-iPSCs (M KSR5) in FBS conditions for up to 10 passages. FBS-to-KSR medium swap caused imprinting instability mostly due to hypomethylation, which was a feature of male KSR-iPSCs. These defects were relatively mild for the M FBS1 iPSC line, but stronger for the M FBS5 iPSC line, with a monoallelic-to-biallelic switch seen for *H19* in both lines and *Snrpn* only in the latter (Supplementary Fig. 7A; Supplementary Data 2). Abnormal hypermethylation was also seen in the maternal allele of the *Dlk1-Dio3* ICR in M FBS1 line, a tendency that was also observed in a few male KSR-iPSCs (Fig. 2A). Overall, these data suggest that imprinting methylation is not correctly maintained in cells cultured in KSR medium, which might explain the hypomethylation defects seen for male KSR-iPSCs. In the reciprocal experiment, KSR-to-FBS medium swap, the original methylation status and allelic expression patterns at imprinted regions were maintained in the M KSR5 iPSC line (Supplementary Fig. 7B; Supplementary Data 2). Allelic expression status of *H19* and *Snrpn* genes were also maintained after KSR-to-FBS swap in a second line, M KSR3 (Supplementary Fig. 7B). FBS medium, therefore, did not induce changes, neither in affected (e.g., *Igf2-H19*)

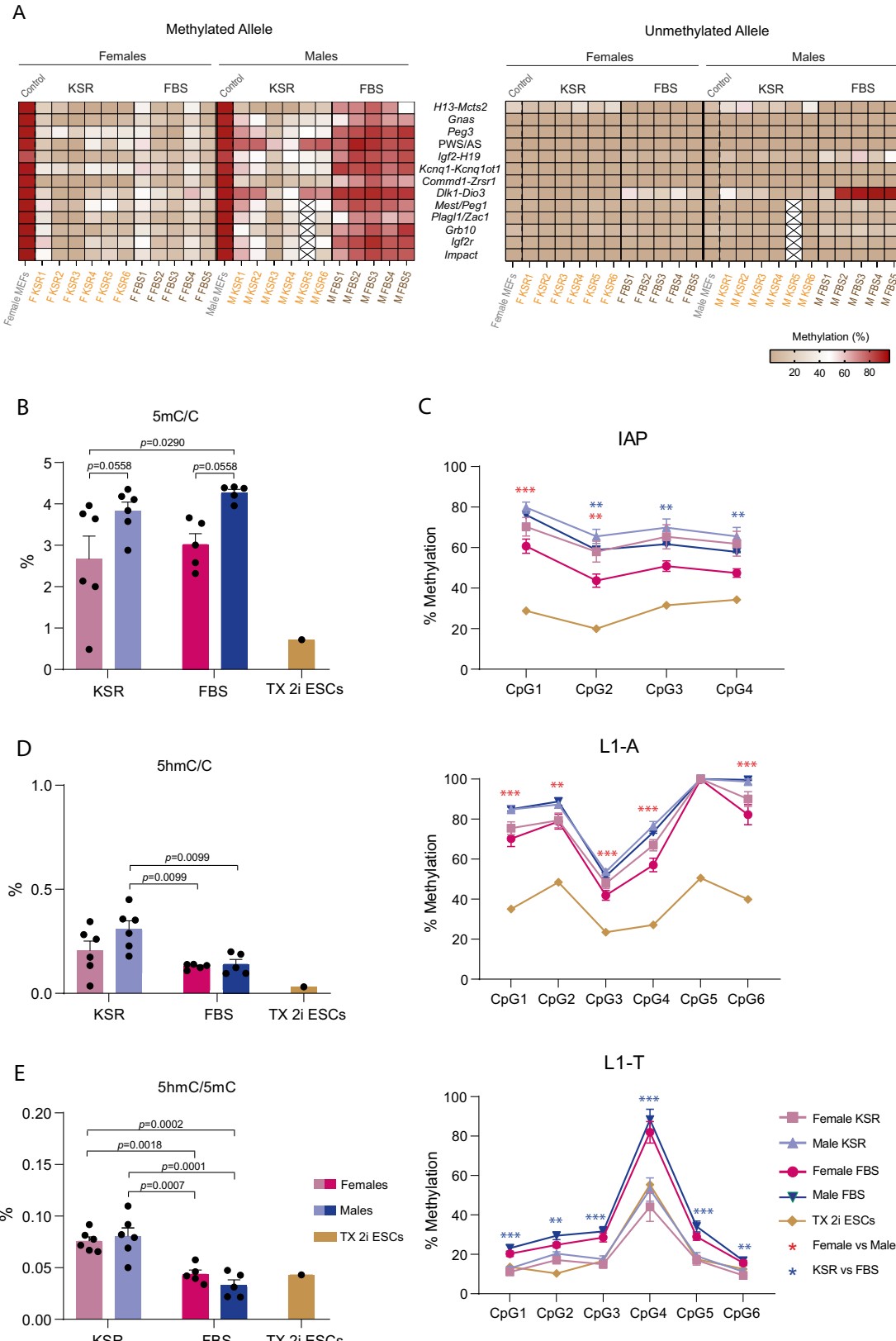

nor in non-affected (e.g., PWS/AS) imprinted regions. This suggests that FBS medium does not lead to a progressive accumulation of defects during culture maintenance within the time-frame explored, nor attenuates or rescues existing defects.

Our data show that KSR-mediated hypomethylation defects at imprinted regions occur independently of the reprogramming process, while female and male iPSCs derived in FBS medium have opposing imprinting defects, hypomethylation in females and hyper-methylation in males (Fig. 5A). Therefore, we next explored when these imprinting defects arise in female and male cells during FBS derivation. Reprogramming intermediates, negative for the fibroblast marker THY1 and positive for the stem cell marker SSEA1 (T−/S+), as well as

**Fig. 5 | Imprinting defects and global methylation changes in female and male KSR-/FBS-derived iPSCs. A** Heatmap representing the percentage of DNA methylation at the methylated (left) and unmethylated (right) alleles of ICRs for female and male MEFs (control) as well as female and male KSR- and FBS-iPSCs; ICRs are shown in rows and samples in columns; boxes with a cross indicate that *Mest/Peg1*, *Plagl1/Zac1*, *Grb10*, *Igf2r* and *Impact* imprinted regions were not assessed for M KSR5 iPSC line. Source data are provided as Supplementary Data 2. **B** Global 5mC levels measured by Liquid Chromatography with tandem mass spectrometry (LC-MS/MS). Graph represents the average percentage ± SEM of 5mC per total cytosines in female and male KSR- and FBS-iPSCs (*n* = 6 and *n* = 5 biologically independent cell lines, respectively), as well as in TX 2i ESCs (*n* = 1); The *p*-values (*p*) comparing female and male KSR- and FBS-iPSCs are indicated on the top of the bars (two-way ANOVA followed by multiple comparisons corrected by the original FDR method of Benjamini and Hochberg). Source data are provided as a Source Data file. **C** Percentage CpG methylation measured by bisulfite pyrosequencing for IAP, L1-A and L1-T repetitive elements. Graph represents the average methylation percentages ± SEM at each CpG sampled per repetitive element in female and male KSR-iPSCs (*n* = 4 and *n* = 5 biological independent cell lines, respectively) and female and male FBS-iPSCs (*n* = 5 biological independent cell lines each) as well as

methylation percentages in TX 2i ESCs (*n* = 1). Statistically significant differences are indicated as ** *p* < 0.01; *** *p* < 0.001 (two-way ANOVA followed by Turkey's multiple comparisons test) with red asterisks meaning statistically significant differences between sex (female vs male) and blue meaning statistically significant differences between medium formulation (KSR vs FBS). Source data are provided as Supplementary Data 2. **D** Global 5hmC levels measured by LC-MS. Graph represents the average percentage ± SEM of 5hmC per total cytosines in female and male KSR- and FBS-iPSCs (*n* = 6 and *n* = 5, biologically independent cell lines each, respectively), as well as in TX 2i ESCs (*n* = 1); The *p*-values (*p*) comparing female and male KSR- and FBS-iPSCs are indicated on the top of the bars (two-way ANOVA followed by multiple comparisons corrected by the original FDR method of Benjamini and Hochberg). Source data are provided as a Source Data file. **E** Ratio of 5hmC/5mC levels measured by LC-MS. Graph represents the average percentage ± SEM of 5hmC per total 5mC in female and male KSR- and FBS-iPSCs (*n* = 6 and *n* = 5, biologically independent cell lines each, respectively), as well as in TX 2i ESCs (*n* = 1); The *p*-values (*p*) comparing female and male KSR- and FBS-iPSCs are indicated on the top of the bars (two-way ANOVA followed by multiple comparisons corrected by the original FDR method of Benjamini and Hochberg). Source data are provided as a Source Data file.

non-reprogramming intermediates (T+/S−) were isolated at day 12 and day 24 by fluorescence-activated cell sorting (FACS)/cell harvesting (Fig. 6A; Supplementary Fig. 7C). As expected, reprogramming intermediates, but not T +/S- sorted cells, express the pluripotency genes, *Nanog* and *Esrrb*, from day 12 at similar level to the established iPSCs (d50 FBS-iPSCs) as shown in Supplementary Fig. 7D for female reprogramming. During reprogramming of female cells, somatic cells transit from a state where one X chromosome is active and another is inactive (XaXi) to a state where both X chromosomes are active (XaXa). To characterise this transition, we measured the RNA levels of *X-inactive-specific-transcript* (*XIST*), the master regulator of XCI, during female reprogramming. We observed a strong reduction of *XIST* levels in reprogramming intermediates at day 12 (Fig. 6B), indicating the existence of the first signs of X-chromosome reactivation. The levels of *XIST* continue to be very low at day 24 and become virtually absent by day 50 (Fig. 6B). This dynamics of *Xist* expression is comparable to previous studies[62,63] and suggest that iPSCs might only be fully established after day 24 of reprogramming.

Then, we monitored global 5mC and 5hmC levels during reprogramming by LC-MS/MS. Global levels of 5mC drop in both female and male day 12-reprogramming intermediates (Supplementary Fig. 7E). This drop is more significant for female cells, possibly due to a combined effect of reprogramming with X-chromosome reactivation, that has been previously linked to global hypomethylation[54,55]. Curiously, after DOX removal, the levels of 5mC initially increase (day 24), but then decrease in fully reprogrammed iPSCs (day 50) for both female and male cells. Importantly, 5mC levels in female cells were always lower than in male cells during reprogramming. A mirror image is observed for 5hmC levels, which show an increase at day 12 (particularly relevant in female reprogramming), decrease at day 24 and increase again at day 50 (Supplementary Fig. 7E). In conclusion, 5mC levels drop at the onset of reprogramming, when the Yamanaka cassette is active, which coincides with an increase in 5hmC levels. This suggests an active demethylation process that, at least in part, might involve TET enzymes. After the silencing of the Yamanaka cassette, 5mC levels go up only to decrease again as the endogenous stem cell program becomes fully operational.

We next monitored imprinting methylation for 6 imprinted regions (*Mcts2-H13*, *Peg3*, PWS/AS, *Igf2-H19*, *Commd1-Zrsr1*, and *Dlk1-Dio3*) and methylated (*Prickle1*) and unmethylated (*Sox2*) controls as well as relative allelic expression levels of *Snrpn* and *H19* genes using RT-PCR followed by Sanger sequencing. While *Sox2* remained unmethylated throughout reprogramming (Supplementary Data 2), *Prickle1* methylation levels closely mimic the global 5mC changes in both female and male cells (Supplementary Fig. 7F). In contrast, DNA

methylation at imprinted clusters did not follow the global 5mC dynamics (Fig. 6C; Supplementary Fig. 7E). In female cells, no hypomethylation errors were observed in imprinted regions at day 12 (Fig. 6C), consistent with the strict monoallelic expression of *Snrpn* and *H19* (Fig. 6D). Hypomethylation defects were however visible at day 24, matching the first signs of derepression of the silent allele of these imprinted genes (Fig. 6C, D). Loss of methylation is further reduced at day 50 when *H19* and *Snrpn* become clearly biallelic (Fig. 6C, D; Fig. 4C). Therefore, loss of imprinting methylation in female iPSCs is uncoupled from the DOX-dependent phase of reprogramming and deviates from the global demethylation seen at this stage. Despite initial resistance to a global decrease in DNA methylation, imprinting methylation is not maintained and is passively lost with time in culture.

In male cells, the first signs of DNA hypermethylation at the *Dlk1-Dio3* locus can be seen as early as day 12 (Fig. 6C) and occur despite a reduction in 5mC and increase in 5hmC levels (Supplementary Fig. 7E). Thanks to the single-molecule resolution of IMPLICON, we could infer that the majority of day 12 reprogramming intermediates present hemi-methylated maternal ICRs, containing both unmethylated and methylated CpGs in the same amplicon. This is eventually resolved in fully methylated (and to a lesser extent fully unmethylated maternal ICRs) (Fig. 6E). Curiously, an increase of DNA methylation in the maternal allele of *Dlk1-Dio3* ICR at day 12 is also observed for non-reprogrammed T+/S− male cells as well as in female reprogramming (Fig. 6C). We postulate that this effect might be directly linked to the activation of the Yamanaka cassette in the first 12 days of reprogramming. Male FBS-iPSCs also show signs of mild hypermethylation at the *Igf2-H19* locus observed from day 24 onwards, without affecting monoallelic expression of *H19* (Fig. 6C; Supplementary Fig. 7G). All the other imprinted regions stay unaffected overall (respectively, > ~60% and < ~10% of methylation levels at the methylated and unmethylated ICRs) until day 50, despite the global changes in 5mC/5hmC during reprogramming (Fig. 6C; Supplementary Fig. 7E). In conclusion, hypermethylation defects in male cells seem to arise during reprogramming under FBS conditions. Adjustments to the protocol are needed to mitigate or correct these defects.

## Strategies to generate male iPSCs without imprinting defects

A key question is whether imprinting defects in iPSCs persist in their differentiated derivatives, as this could have major implications for their translational and clinical applications. To test this, we differentiated iPSCs with hypomethylation (F FBS5) and hypermethylation (M FBS5) defects into neural progenitor cells (NPCs) (Supplementary Fig. 8A). IMPLICON revealed the persistence of both hypo- and

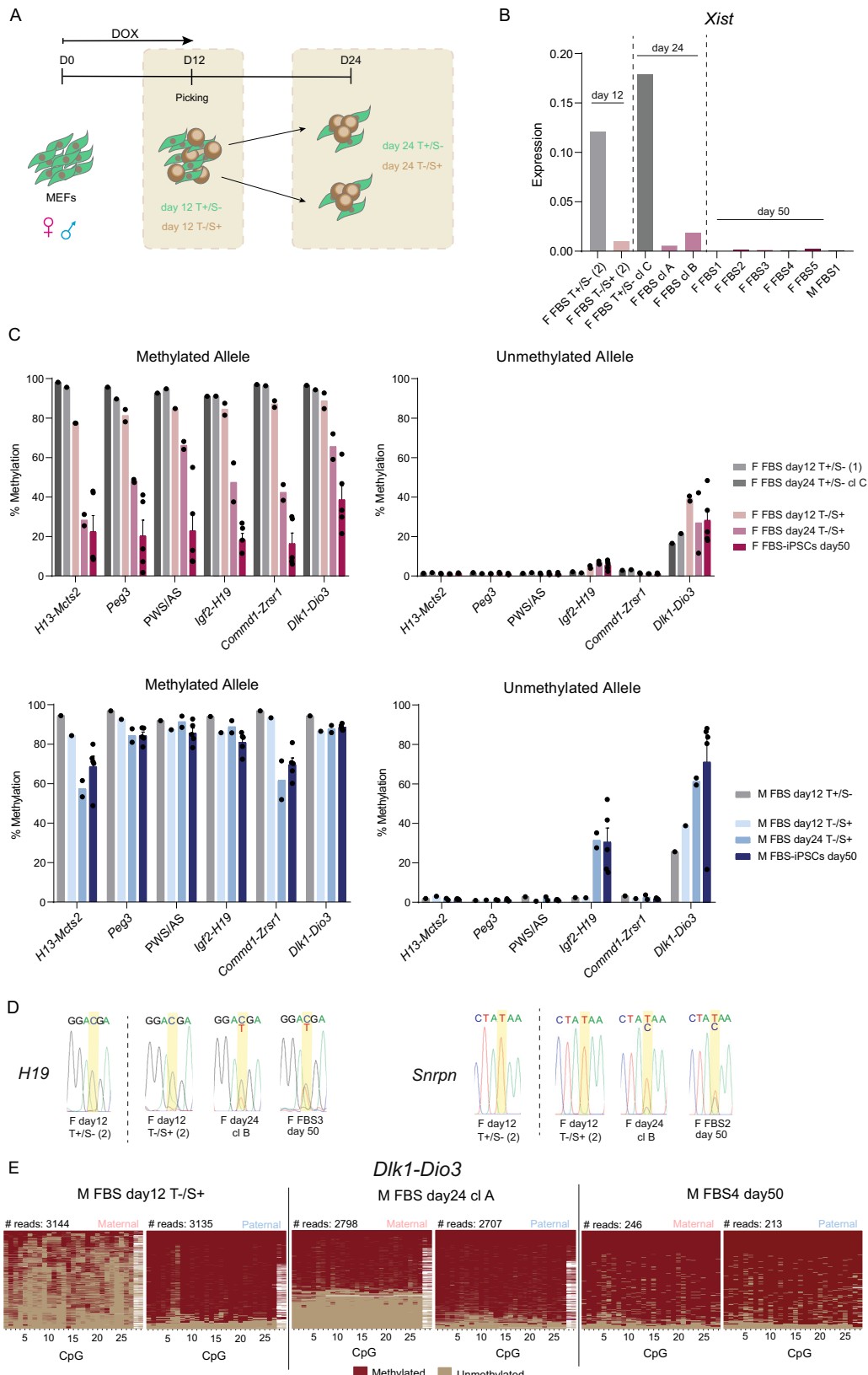

hypermethylation errors and abnormal expression of imprinted genes in the corresponding NPC lines (Supplementary Fig. 8B, C). In contrast, the *Prickle1* non-imprinted gene recovered the methylation levels seen in MEFs (Supplementary Fig. 8B). While different genomic regions adjust their methylation levels in response to environmental cues (reprogramming/stem cell maintenance/differentiation), abnormal

changes in the epigenetic status of imprinted regions persevere, which can have long-lasting consequences, as previously shown[17,35].

In contrast to female iPSCs and male KSR-iPSCs, imprinting errors in male FBS-iPSCs were induced by reprogramming. Thus, we decided to explore two strategies to correct these reprogramming-induced imprinting errors. Our first approach consisted in a medium

**Fig. 6 | Emergence of imprinting defects in male and female FBS-iPSCs during reprogramming. A** Schematic representation of the time-points for collection of reprogramming intermediates; briefly, F1 hybrid mouse embryonic fibroblasts (MEFs) containing the Yamanaka cassette were reprogrammed in Foetal Bovine Serum medium (FBS) in the presence of doxycycline (DOX) for 12 days. THY1-/SSEA1+ (T−/S+) reprogramming intermediates and THY1+/SSEA1- (T+/S-) non-reprogramming intermediates were FACS sorted at day 12 and day 24. In the case of iPSC clones already free of MEFs at day 24, iPSCs were harvested with no need for cell sorting (see Methods). **B** RT-qPCR expression analysis of *Xist* expression normalised with the *Gapdh* housekeeping gene in female cells collected during reprogramming (day 12, 24 and 50); each bar represents data from only one biological replicate. T+/S- non-reprogramming intermediates; T−/S+ reprogramming intermediates. Source data are provided as a Source Data file. **C** Average percentage of methylation at methylated and unmethylated alleles of ICRs in female and male cells sorted/collected at day 12 (T+/S−, $n = 1$ for both biological sexes; female T−/S+ ,

$n = 2$; male T−/S+ , $n = 1$) and at day 24 (female T+/S-, $n = 1$; T−/S+ , $n = 2$ for both biological sexes), as well day 50 fully reprogrammed iPSCs (note: same data as in Supplementary Fig. 4E for FBS-iPSCs day 50, $n = 5$ for both biological sexes); Graph represents the mean ± SEM methylation levels measured at each CpG within different genomic regions per parental allele for each group of samples; Source data are provided as Supplementary Data 2. **D** Allelic expression levels of *H19* and *Snrpn* assayed by RT-PCR followed by Sanger sequencing during female reprogramming. Chromatograms of *H19* gene are shown for F day12 T+/S- (2), F day12 T−/S+ (2), F day24 cl B as well as for fully reprogrammed F FBS3 collected at day 50. Chromatograms for *Snrpn* gene are shown for F day12 T+/S- (2), F day12 T−/S+ (2), F day24 cl B as well as for fully reprogrammed F FBS2 collected at day 50. E. Plots display methylated and unmethylated CpGs for each CpG position (in columns) in all the individual reads (in rows) for both maternal and paternal alleles of *Dlk1-Dio3* imprinted locus in the reprogramming intermediates, M FBS day 12 T−/S+ and M FBS day 24 clone A, as well as in fully reprogrammed M FBS4 iPSCs collected at day 50.

---

formulation containing KSR and FBS in a 1:1 ratio, reasoning that hypomethylation defects induced by KSR could be offset by FBS, while the hypermethylation induced by FBS could be rescued by KSR. This approach fully rescued the hypermethylation defects of the unmethylated allele of *Igf2-H19* locus and substantially improved the ones on the *Dlk1-Dio3* region in 4 out of 6 KSR/FBS-iPSC lines (Fig. 7A). However, we could also detect additional hypomethylation defects at the methylated allele of several loci (*Mcts2-H13*, *Peg3*, *Igf2-H19* and *Commd1-Zrsr1*) (Fig. 7A), accompanied by biallelic expression of imprinted genes (Supplementary Fig. 9A). Despite some improvements, the KSR/FBS formulation still induces hypomethylation imprinting defects previously seen in KSR-iPSCs.

Our second approach consisted in adding VitC to the FBS medium. FBS+VitC formulation was previously shown to strongly improve the hypermethylation phenotype at the *Dlk1-Dio3* ICR[38], but its impact on other imprinted regions has not been systematically investigated. Under our reprogramming conditions, we observed no imprinting errors in *Peg3*, PWS/AS and *Commd1-Zrsr1* loci, associated with correct *Snrpn* monoallelic expression (Fig. 7A; Supplementary Fig. 9A). We detected a mild decrease in DNA methylation of *H13-Mcts2* and mild hypermethylation of the *Igf2-H19* cluster, but this did not affect *H19* monoallelic expression pattern (Fig. 7A; Supplementary Fig. 9A). Overall, the results matched the ones obtained under FBS conditions, with VitC addition not impacting negatively in most loci. In the case of *Dlk1-Dio3*, we saw that addition of VitC partially corrected the hypermethylation defects in 3 out of 6 FBS+VitC-iPSCs (Fig. 7A). We further investigated these results at single-molecule resolution and observed that the maternal ICR exists in two forms: fully unmethylated or fully methylated (Fig. 7B). This indicates that these iPSC lines have a mixed population where some cells exhibit a normal imprinting pattern, while others exhibit a hypermethylated phenotype. In order to get iPSCs with normal *Dlk1-Dio3* imprinting, we isolated single-cell clones from M FBS+VitC5 and measured *Meg3* expression levels. We were able to isolate 1 subclone (cl5) expressing normal *Meg3* levels indicating that iPSCs with normal imprinting at the *Dlk1-Dio3* locus could be successfully obtained (Fig. 7C). Interestingly, this subclone also expresses *H19* levels comparable to the levels in M FBS2 which has normal imprinting at *Igf2-H19* locus (Fig. 4B; Fig. 7A; Supplementary Fig. 9B). Our results show that FBS+VitC medium formulation improved imprinting fidelity in mouse iPSCs. These data are strongly supporting the possibility of generating epigenetically faithful iPSCs, following further optimizations/adaptations of the current reprogramming protocols.

## Discussion

In this study, we thoroughly analysed the fidelity of genomic imprinting in mouse iPSCs. By using an established reprogramming system[40,41] combined with an ultra-deep approach to screen imprinting methylation on murine hybrid cells, we explored the impact of the sex

of the donor cell and reprogramming culture conditions on the epigenetic stability of imprinted loci. Our comprehensive analysis resulted in several important findings: (1) female sex is a strong predictor of hypomethylation imprinting errors in iPSCs; (2) KSR-based medium induces hypomethylation defects at many ICRs; (3) accumulation of imprinting errors is not a mere consequence of DNA methylation fluctuations during reprogramming; (4) classical FBS-based ESC medium conserves correct imprinting in many ICRs in male cells, but renders paternal methylated ICRs prone to gain methylation on the maternal allele during reprogramming; (5) KSR/FBS formulation recovers the hypermethylated phenotype to a great extent, but at the cost of hypomethylation defects at multiple imprinted loci; (6) Addition of VitC to FBS medium has a positive effect on *Dlk1-Dio3* locus without affecting imprinting at other loci. Our thorough analysis provides an explanation for the contrasting results from previous studies[24,27–29], when sex of the donor cell (when available) and reprogramming culture conditions are considered (Supplementary Table 1). This knowledge is important to choose the best reprogramming strategies to create iPSCs devoid of imprinting abnormalities.

The allele-specific IMPLICON creates a final dataset reflecting the original methylated/unmethylated DNA molecules with nucleotide and allelic resolution[44]. This technique allowed us to investigate, for the first time, the heterogeneity of epigenetic states at imprinted regions within iPSC clonal lines. Indeed, we could show that partial hypomethylation defects for certain loci (e.g., *Kcnq1-Kcnq1ot1*, *Gnas*) were due to the existence of two major classes of DNA molecules, one with an intact ICR (with a methylated and an unmethylated allele) and another with complete erasure of the methylation marks at this element (both alleles unmethylated) (Supplementary Fig. 2C). This suggests that within an iPSC line, a subset of cells retained proper methylation status for a given locus, while the rest lost methylation. Our IMPLICON datasets were also fundamental to recover cells with normal imprinting at the *Dlk1-Dio3* and *Igf2-H19* loci in one FBS+VitC-iPSC line with heterogeneous epigenetic states (Fig. 7B, C; Supplementary Fig. 9B). Our data points out for intra-heterogeneity of imprinted defects at certain loci within clonal lines which was not previously appreciated.

Female sex is the major predictor of hypomethylation defects at imprinted loci in mouse iPSCs (Fig. 5A). This is, however uncoupled from the fluctuations of DNA methylation during reprogramming. Early in reprogramming, imprints are maintained and monoallelic expression of imprinted genes is unaffected, despite an accentuated decrease in global DNA methylation levels that is accompanied by a peak in 5hmC levels (Fig. 6C; Supplementary Fig. 7E). These trends in global 5mC/5hmC levels are also present more modestly during male reprogramming (Supplementary Fig. 7E) suggesting that this is a feature of the reprogramming process. This was well documented for 5mC levels[57], but to our knowledge we are documenting the dynamics of 5hmC levels during reprogramming for the first time. Importantly,

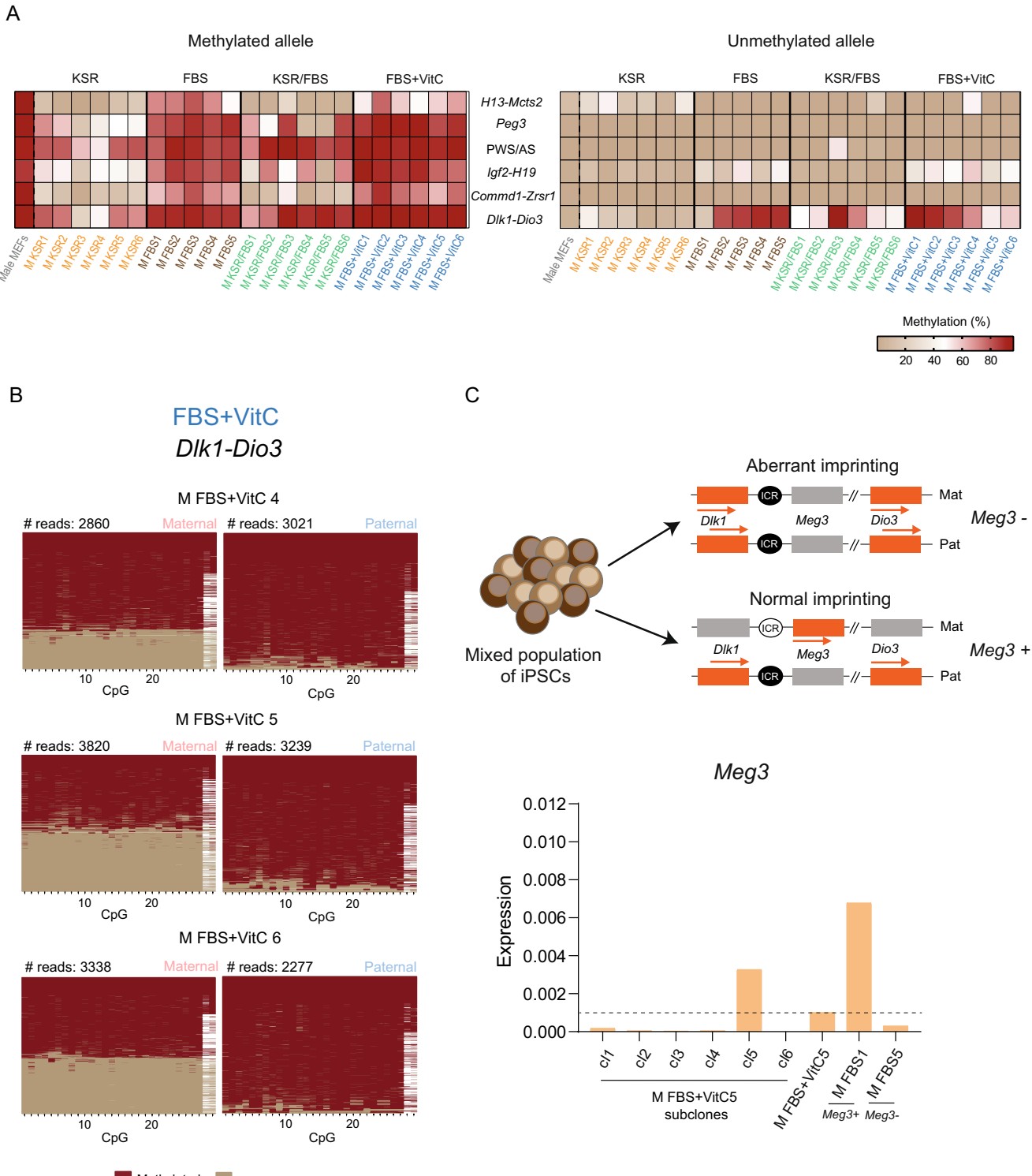

**Fig. 7 | Imprinting fidelity in male KSR/FBS- and FBS + VitC-iPSCs. A** Heatmap representing the percentage of DNA methylation at the methylated (left) and unmethylated (right) alleles of ICRs for male MEFs as well as male KSR- and FBS-iPSCs, KSR/FBS-iPSCs and FBS + VitC-iPSCs (note: same data as in Fig. 5A for male MEFs, KSR- and FBS-iPSCs). ICRs are shown in rows and samples in columns. Source data are provided as Supplementary Data 2. **B** Plots display methylated and unmethylated CpGs for each CpG position (in columns) in all the individual reads (in rows) for both maternal and paternal alleles of *Dlk1-Dio3* imprinted locus in M FBS+VitC4-6 iPSCs. **C** Expression analysis of *Meg3* gene in subclones of the FBS + VitC5 iPSCs. On top, the scheme represents M FBS+VitC5 iPSCs with a mixed population of cells with normal and abnormal imprinting as estimated by the

results in Fig. 7B. The status of *Dlk1-Dio3* ICR with normal and abnormal (hypermethylation) imprinting are also displayed (white circle – unmethylated ICR; black circle – methylated ICR; Mat – maternal allele; Pat – paternal allele; orange rectangles – expressed genes; grey rectangles – silenced genes; regions are not drawn to scale). On the bottom, graph represents the RT-qPCR expression analysis of *Meg3* expression normalised with the *Gapdh* housekeeping gene in the subclones of M FBS+VitC5, original M FBS+VitC5 and M FBS1 (normal imprinting) and M FBS5 (hypermethylated phenotype); dashed line represents the *Meg3* expression level of the parental M FBS+VitC5 iPSC line; each bar represents data from only one biological replicate. Source data are provided as a Source Data file.

the methylation pattern of the non-imprinted gene *Prickle1*, follows these 5mC fluctuations, suggesting that imprinting defects are not just a reflection of overall DNA methylation dynamics (Supplementary Fig. 7F). This decrease in 5mC levels is exacerbated in female cells because these cells downregulate *Xist*, a sign of initiation of X-chromosome reactivation (Fig. 6B). The 5hmC peak in females (Supplementary Fig. 7E) suggests that active demethylation through TET enzymes is at play and presumably is enhanced due to X-chromosome reactivation, a link that has not been previously made. Hypomethylation at imprinted regions slowly arises at intermediate stages, leading to complete loss of methylation and biallelic expression of imprinted genes only in fully reprogrammed iPSCs (Fig. 6C; Fig. 6D). In this time-frame, 5mC levels initially go up, possible due to DOX withdrawal and inactivation of the Yamanaka cassette, and then go down presumably due to full activation of the endogenous pluripotent program that guarantees full repression of *Xist* gene (Supplementary Fig. 7E; Fig. 6B). Through all these stages DNA methylation levels in females remain lower than male cells (Supplementary Fig. 7E), leading to passive loss of imprinting. In this respect, they are not different from what was observed in female ESCs[54]. Higher dosage of X-linked genes due to the presence of two active Xchromosomes have been implicated in this phenotype. In particular, the X-linked *Dusp9* gene has been implicated in hypomethylation of female stem cells[64], however, other X-linked genes might also be involved during reprogramming[65]. As X-chromosome reactivation is a hallmark of female iPSCs, strategies to protect imprints by manipulating gene expression from X-linked genes need to be envisioned. Alternatively, as epiblast stem cells (EpiSCs) retain an inactive X-chromosome, the generation of induced EpiSCs (iEpiSCs)[66] might provide an alternative to produce female pluripotent stem cells devoid of imprinting defects. A thorough analysis on X-chromosome status during iEpiSC derivation will provide insights about the feasibility of this strategy.

KSR-based medium was also a driver of hypomethylation defects at ICRs, even in male iPSCs. Our FBS-to-KSR swap experiments revealed that culturing iPSCs in KSR medium for at least 10 passages induces hypomethylation defects (Supplementary Fig. 7A). Therefore, this serum-free formulation causes imprinting errors. In fact, this KSR effect was also found in the context of the KSR/FBS 1:1 formulation (Fig. 7A). Imprinting defects in KSR medium occur even though overall 5mC levels were largely untouched and common repetitive elements were still methylated (with the exception of L1-T element) (Fig. 5B, C). Hypomethylation at ICRs correlated with elevated *Tet1*/*Tet2* expression levels (Supplementary Fig. 6) and increase of 5hmC levels globally (Fig. 5D). The effect on 5hmC levels might also be enhanced by the presence of VitC in the KSR formulation[38] that promotes the activity of TET dioxygenase enzymes and reduces DNA methylation levels[39] as we discuss below.

Male FBS-iPSCs have mild and strong hypermethylation defects in the paternally methylated *Igf2-H19* and *Dlk1-Dio3* loci, respectively. Recently, it has been documented that the *Dlk1-Dio3* locus is more sensitive to hypermethylation defects on pluripotent stem cells of the BL6 mouse strain[67], which is the genetic background of the maternal locus in our hybrid BL6/CAST iPSCs. Our results match the ones observed by other authors using a different hybrid cross, 129×1/SvJ × MSM/Ms, for both *Igf2-H19* and *Dlk1-Dio3* loci[29], showing that these intergenic paternally methylated ICRs are particularly susceptible to reprogramming-induced errors in different genetic backgrounds. Imprinting defects are first detected at day 12 for the *Dlk1-Dio3* (Fig. 6C). Thanks to the single-molecule resolution of IMPLICON, we could see that the majority of male reprogramming cells have a variegated maternal ICR with both methylated and unmethylated CpGs (Fig. 6E). Strikingly, this hypermethylation at *Dlk1-Dio3* locus is also observed for female cells and non-reprogramming intermediates, suggesting a direct link with the activation of the Yamanaka cassette (12 days in our protocol), despite the global decrease in DNA

methylation. Regarding *Igf2-H19* locus, the defects occur between day 12 and 24 and then are kept unaltered until day 50 in fully reprogrammed iPSCs. Hypermethylation defects in this locus are fully recovered with the KSR/FBS 1:1 formulation, which are also alleviated at the *Dlk1-Dio3* locus. However, the success of the KSR/FBS formulation was offset by undesirable hypomethylation defects in other regions, including *Igf2-H19* locus (Fig. 7A). Given that VitC in the KSR formulation impacts TET function and affects DNA methylation, our preferred reprogramming strategy relies on the addition of VitC to the FBS medium. We used a concentration of VitC similar to the one used previously[38], who showed marked recovery of hypermethylation defects in the *Dlk1-Dio3* cluster. We avoided higher concentrations previously used to significantly increase 5hmC levels in stem cells[39] to reduce the potential negative impact on other imprinted regions, given the outcome with the KSR and KSR/FBS formulations. Under our conditions, VitC did not have a negative impact in other imprinted regions and the mild hypermethylation phenotype at the *Igf2-H19* locus still persisted. Importantly, an improvement in the hypermethylation phenotype was observed in 3 out of 6 FBS+VitC-iPSCs (Fig. 7A). These iPSCs exhibit a mixed population of cells with normal and abnormal imprinting that were then separated by subcloning (Fig. 7C). Despite this improvement, our results at the *Dlk1-Dio3* locus were less impressive compared to the previous attempt[38]. Differences in the genetic background or other technical aspects (e.g., different FBS batches) might have contributed to these differences. Playing with higher concentrations of VitC is an avenue to explore in the future, bearing in mind that any improvement in hypermethylation defects at the *Dlk1-Dio3* or *Igf2-H19* could be offset by hypomethylation defects in other imprinted loci. In conjunction to VitC adjustments, alternative reprogramming setups could also be considered such as, the use of different stoichiometries of the 4 Yamanaka factors or even the exclusion of the *Oct4* from the Yamanaka cocktail which may have a positive impact on imprinting maintenance[36,68].

Our study clearly shows frequent imprinting errors in iPSCs. These imprinting defects are known to cause developmental/metabolic phenotypes in mice and cause imprinting disorders in humans[30,69]. These phenotypes highlight a clear negative impact of using iPSCs for modelling imprinting disorders. But whether defective imprinting could majorly affect the downstream applications of iPSCs in disease modelling, drug toxicology, compounds screening or cell therapy remain a virgin ground to explore. Our pluripotency assay based on teratoma formation showed that iPSCs with either hypo- or hypermethylation defects (including at the *Dlk1-Dio3* region) could readily generate well-differentiated teratomas. Generation of chimeric mice or all-iPSC mice through blastocyst injections would provide a more refined way to address the developmental potential of distinct imprinting defects that result from the reprogramming of female and male iPSCs in different conditions and should be explored in the future in a controlled manner. In any case, based on the teratoma results, we would anticipate that imprinting dysregulation in iPSCs does not impede the major axes of lineage commitment. Imprinting defects might rather affect specific cellular functions or differentiation of defined cellular subtypes, as it has been observed for the role of imprinted gene *IGF2* in hematopoietic commitment[10] or *MEG3* in neural differentiation[56].

Although neglected as a hallmark of pluripotency[70], imprinting fidelity was brought back to spotlight by the existence of multiple imprinting errors in cells derived from the first human trial of iPSC-based autologous transplantation[71]. Indeed, human iPSCs are also known to have frequent imprinting defects, notably at the *DLK1-DIO3* region[16,17,35]. Moreover, many human iPSCs have been derived using KSR as a component of the medium, which in mouse stem cells negatively impacts imprinting maintenance (Fig. 5A; Fig. 7A; Supplementary Fig. 7A). In any case, we should be cautious to directly translate mouse iPSC data to human cells due to the multiple

differences in medium requirements and epigenetic states between mouse and human iPSCs. For instance, human female iPSCs do not undergo inactive X-chromosome reactivation during reprogramming[72] and, thus, female sex may not have a major impact on imprinting defects during reprogramming of human iPSCs. Yet, many human female iPSCs/ESCs partially reactivate the inactive X upon prolonged in vitro culturing[25,26] and the consequences of this on imprinting are unknown. All in all, our findings in murine iPSCs raise awareness about the importance of testing human iPSCs (or ESCs) for imprinting fidelity using methods such as IMPLICON which can also be used to assess human imprints[44]. Imprinting screening should be part of a quality control panel to assure safety for the use of human iPSCs for disease modelling approaches, and more importantly, for clinical applications.

## Methods

### Ethics
Animal care and experimental procedures (see MEFs generation and teratoma assay sections) involving mice were carried out in accordance with European Directive 2010/63/EU, transposed to the Portuguese legislation through DL 113/2013. The use of animals has been approved by the Animal Ethics Committee of IMM JLA and by the Portuguese competent authority – Direcção Geral de Alimentação e Veterinária – with licence numbers 015229/17 and 023357/19.

### Mice strains
The "*reprogrammable*" transgenic i4F-BL6 (*Mus musculus*, C57BL/6J, kind gift from Manuel Serrano, IRB, Barcelona)[40], CAST (*Mus musculus castaneus*, CAST/EiJ, the Jackson Laboratory) and NSG mice (*Mus muslucus*, NOD.Cg- Prkdcscid Il2rgtm1Wjl/SzJ, the Jackson Laboratory)[47] colonies were maintained at the Instituto de Medicina Molecular João Lobo Antunes (iMM JLA) Rodent facility. The i4F-BL6 mouse colony was always maintained on the BL6 background during this study, except when crossed with CAST animals to obtain F1 embryos (see Generation and maintenance of F1 hybrid MEFs). Animals were housed in a maximum of five per cage in a temperature- and humidity-controlled room (24 °C, 45–65%) with a 14/12 hr light/dark cycle. Animals were fed diet *ad libitum*.

### Generation and maintenance of F1 hybrid MEFs
Crosses between 2 female 8-week-old i4F-BL6 and 2 male 12-week-old CAST were set up to generate F1 hybrid transgenic E13.5 mouse embryos. MEFs from 3 individual embryos (two males and one female) were collected as previously described[73]. Briefly, individual embryos are removed from the uterus and dissected in Phosphate-buffered saline (PBS) and Penicillin-Streptomycin (P/S; Cat# 15140-122, Gibco). Mashed tissue is initially incubated with 2% Trypsin-EDTA 1X (Cat# 25300-062, Gibco) at 37 °C. A Pasteur pipette is used to disperse the suspension of cells that were then incubated in Dulbecco's modified Eagle's medium (DMEM; Cat# 10569-010, Gibco) containing 10% FBS (Cat# 10270-106, Gibco), 2 mM GlutaMAX (Cat# 35050-061, Gibco) and 100 μg/ml P/S. MEFs were expanded and frozen at passage 2 (P2) prior to the reprogramming experiments. Genotyping for the inducible Yamanaka cassette and the *rtTA* gene in the *Rosa26* locus was performed using the primers in Supplementary Table 2.

### Reprogramming of MEFs
F1 hybrid MEFs at P2 were plated and expanded until reaching confluency of 40% per well in a gelatin-coated 6 well plate and cultured in KSR, FBS, KSR/FBS or FBS+VitC culture conditions (see below) with 1.5 μg/ml of DOX (Cat# D9891, Sigma-Aldrich). The medium was changed every 48 h. Individual iPSCs colonies were picked at day 12 after DOX induction using glass cloning cylinders (Sigma-Aldrich) and 0.05% Trypsin-EDTA 1X (Cat# 25300-054, ThermoFisher Scientific) and plated into a previously gelatin-coated 96-well plate on feeders without DOX. Each iPSC colony was then transferred subsequently into 24,

12 and 6-well plates and 25 cm² culture flasks. Each iPSC was then continuously passed for 2–4 passages until reaching approximately day 50 after initiation of reprogramming. All cells and reprogramming experiments were performed in an incubator at 37 °C and 5% CO₂ in normoxia conditions. The same batch of female and male MEFs derived from the single embryos of the same progeny were used to generate KSR-iPSCs and FBS-iPSCs. Another batch of male hybrid MEFs from another progeny were used to generate KSR/FBS-iPSCs and FBS+VitC-iPSCs.

### Stem cell culture conditions
KSR-iPSCs were cultured in high-glucose DMEM supplemented with 15% KSR (Cat# 10828-028, Gibco), LIF (1000 U/ ml; Cat# ESG1107, Millipore), 1% MEM non-essential amino acids (Cat# 11140-050, Gibco), 0.5% P/S (Cat# 15140-122, Gibco), 1% glutamax (Cat# 35050-061, Gibco) and 0.2% β-mercaptoethanol (Cat# 31350-010, Gibco). FBS-iPSCs and JM8.F6 ESCs[45] (kind gift from Manuel Serrano, IRB, Barcelona) were cultured in standard serum medium conditions which contains high-glucose DMEM supplemented with FBS (15%; Cat# 16141-079, Gibco), LIF (1000 U/ ml), 0.5% P/S, 1% glutamax, and 0.2% β-mercaptoethanol. TX1072 (TX 2i) ESCs[46,74] were cultured in the same medium conditions with the addition of the 2i chemical inhibitors: 3 μM of CHIR99021 (Cat# SML1046, Sigma) and 250 μM of PD0325901 (Cat# PZ0162, Sigma). FBS + VitC-iPSCs were cultured in FBS-iPSCs medium (described above) with the addition of 0.5 μM of ascorbic acid (Cat# A4544, Sigma). KSR/FBS-iPSCs were cultured in high-glucose DMEM supplemented with 10% KSR and 10% FBS, LIF (1000 U/ ml), 0.5% MEM non-essential amino acids, 0.5% P/S, 1% glutamax, and 0.2% β-mercaptoethanol. For the swap experiments, M FBS1 and M FBS5 iPSC lines were cultured in KSR formulation, while M KSR3 and M KSR5 iPSC lines were cultured in FBS formulation for 10 passages (~ 20 days).

### Fluorescence-Activated Cell Sorting (FACS) during reprogramming
To assess imprinting methylation during reprogramming, female and male MEFs were induced with DOX for 12 days and processed for FACS analysis (see below) or picked as single clones and cultured until day 24. On day 24, only iPSC clones still presenting MEFs were FACS-sorted, while the other iPSC clones were directly harvested for DNA extraction.

For FACS, T+/S− and T−/S+ female and male cells under reprogramming in FBS culture conditions were sorted based on the expression of the antibodies 1:100 of PE anti-mouse CD90.2/Thy1.2 (Biolegend, clone 30-H12, Cat#105307) and 1:200 of Brilliant Violet 421 anti-mouse CD15/SSEA-1 (Biolegend, clone MC-480, Cat# 125613) using a gating strategy exemplified in Supplementary Fig. 10. Incubation was performed at 4 °C for 30 min and then stained cells were resuspended in PBS + 2% FBS and transferred to cytometry tubes and sorted using a BD FACSAria IIu (BD Biosciences). Purity of the sorted samples was checked after sorting and data were analysed using FlowJo software v10.7.2. The samples sorted by FACS are named as: F FBS day12 T+/S− (1), F FBS day12 T−/S+ (1), F FBS day12 T+/S− (2), F FBS day12 T−/S+ (2), F FBS day24 T+/S− cl C, F FBS day24 T−/S+ cl C, M FBS day12 T+/S− and M FBS day12 T−/S+. The samples harvested directly for DNA extraction are named as: F FBS day24 cl A, F FBS day24 cl B, M FBS day24 cl A and M FBS day24 cl B. All Flow Cytometry experiments were performed at the Flow Cytometry Facility of Instituto de Medicina Molecular João Lobo Antunes, Lisboa, Portugal.

### Teratoma assay
To evaluate the capacity for teratoma formation, KSR- and FBS-iPSCs (F KSR2, F KSR4, M KSR3, M KSR5, F FBS1, M FBS1 and M FBS5) lines were trypsinized and $2 \times 10^6$ cells were subcutaneously injected into the flanks of 3-month-old immunocompromised male NSG mice (7 animals in total, one each per cell line). Animals were sacrificed with

anaesthetic overdose and a necropsy was performed. Subcutaneous tumours were harvested, fixed in 10% neutral-buffered formalin, stained with hematoxylin and eosin and examined by a pathologist blinded to experimental groups using a Leica DM2500 microscope coupled to a Leica MC170 HD microscope camera.

## Immunofluorescence (IF)

Cells previously seeded on gelatin-coated coverslips were fixed with 3% paraformaldehyde for 10 min at room temperature, washed with PBS and permeabilized in 0.5% Triton X-100 in PBS for 4 minutes (min) on ice. A blocking step was performed by incubation with 1% Bovine Serum Albumin (BSA; Cat# 05470-5 G, Sigma-Aldrich) in PBS for 15 min and subsequently the cells were incubated with primary antibodies for OCT4 (monoclonal clone 7F9.2 m, Cat# MAB4419, Merck Millipore, 1:200 dilution), SSEA-1 (monoclonal clone MC-480, Cat# MAB4301, Merck Millipore, 1:100 dilution) and NANOG (polyclonal, Cat# RCAB002P-F, Reprocell, 1:150 dilution) diluted in 1% BSA/PBS for 45 min. After three washes with PBS, cells were incubated for 45 min with the secondary antibody Cy™3 AffiniPure F(ab')₂ Fragment Goat Anti-Mouse IgG (H + L) (polyclonal, Cat# 115-166-003, Jackson ImmunoResearch Laboratories Inc., 1:200 dilution). 40,6-diamidino-2-phenylindole (DAPI, 0.2 mg/ml; Cat# D9542, Sigma) was used to stain the DNA and mark the nuclei by incubating at RT for 2 min. Cells were imaged using Zeiss Axio Observer (Carl Zeiss MicroImaging) with 63× oil objective using the filter sets FS43HE and FS49 and digital images were processed using FIJI platform (ImageJ v2.1.0/1.53q; [https://fiji.sc/]).

## Karyotyping

iPSCs were treated with colcemid (0.5 μg/mL; Cat# 15212-012, ThermoFisher) for 4 hours at 37 °C to arrest cells in metaphase. Cells were then harvested by trypsin incubation during 5 min at 37 °C, treated with hypotonic potassium chloride solution for 30 min at 37 °C and resuspended and fixed in glacial acetic acid:methanol (1:3). Cells were dropped onto slides and stained with DAPI for 5 mins and observed using Zeiss Axio Observer (Carl Zeiss MicroImaging) with 63× oil objective. An average of 25 metaphase spreads were counted per each of the 8 iPSC lines (M KSR3, M KSR5, F KSR2, F KSR4, M FBS1, M FBS5, F FBS1 and F FBS5), making a total of 200 metaphases analysed. A normal karyotype was considered when most of the metaphases counted presented 40 chromosomes.

## NPC differentiation

The F FBS5 and M FBS5 iPSCs lines were differentiated into neural progenitor cells (NPCs) as previously described[75,76]. Briefly 1 × 10⁶ cells were plated on 0.1% gelatin-coated dishes in N2B27 medium (DMEM/F12 (Cat# 31330-038): Neurobasal [1:1] (Cat# 21103-049), supplemented with 1X L-Glutamine (Cat# 25030-024), 0.5X B27 (Cat# 17504-044), 0.1 mM 2-mercaptoethanol (Cat# 31350-010), all from Thermofisher, and 1X N2 (Cat# SCM012, Millipore) and grown for 7 days. Cells were then trypsinized and 3 × 10⁶ cells were plated on non-adherent Petri dishes to induce neurospheres formation in N2B27 medium supplemented with 10 ng/ml of both EGF (Cat# 315-09, Peprotech) and FGF2 (Cat# 100-18B, Peprotech). After 3 days, neurospheres were allowed to attach on gelatin-coated tissue culture dishes in the same medium, so that NPC could expand from the attached spheres. NPC lines were thereafter maintained in N2B27 medium supplemented with EGF and FGF2 (10 ng/ml each), on 0.1% gelatin-coated flasks.

## DNA extraction and bisulfite treatment

Genomic DNA from parental female and male MEFs, respective KSR-, FBS-, KSR/FBS- and, FBS+VitC-iPSCs, TX 2i ESCs as well as female and male reprogramming intermediates/non-reprogramming intermediates at days 12/24, were isolated using conventional phenol:chloroform:isoamyl alcohol extraction. 1 μg of genomic DNA was

bisulfite converted using the EZ DNA methylation Gold kit (Cat# D5006, Zymo Research) according to manufacturer's instructions and eluted, after column cleanup, in an elution buffer (66 μl) to obtain a final concentration of ~15 ng/μl bisulfite converted DNA.

## 5mC/5hmC measurements by liquid chromatography-mass spectrometry

Genomic DNA from both female and male KSR- and FBS-iPSCs, TX 2i ESCs and female and male reprogramming intermediates/non-reprogramming intermediates (day12 and 24) was digested using DNA Degradase Plus (Zymo Research, Cat# E2020) according to the manufacturer's instructions. Nucleosides were analysed by LC-MS/MS on a Q-Exactive mass spectrometer (Thermo Scientific) fitted with a nanoelectrospray ion-source (Proxeon). All samples and standards had a heavy isotope-labelled nucleoside mix added prior to mass spectral analysis (2′-deoxycytidine-¹³C₁, ¹⁵N₂ (Santa Cruz, Cat# SC-214045), 5-(methyl–²H₃)-2′-deoxycytidine (Santa Cruz, Cat# SC-217100), 5-(hydroxymethyl)–2′-deoxycytidine-²H₃ (Toronto Research Chemicals, Cat# H946632). MS2 data for 5hmC, 5mC and C were acquired with both the endogenous and corresponding heavy-labelled nucleoside parent ions simultaneously selected for fragmentation using a 5 Th isolation window with a 1.5 Th offset. Parent ions were fragmented by Higher-energy Collisional Dissociation (HCD) with a relative collision energy of 10%, and a resolution setting of 70,000 for MS2 spectra. Peak areas from extracted ion chromatograms of the relevant fragment ions, relative to their corresponding heavy isotope-labelled internal standards, were quantified against a six-point serial 2-fold dilution calibration curve, with triplicate runs for all samples and standards.

## Bisulfite Pyrosequencing

Bisulfite-converted DNA was used for PCR amplification using specific primers for L1-A, L1-T and IAP repetitive elements in a 25 μL reaction volume containing 0.4 μM forward and reverse primers, KAPA HiFi HotStart 2X (Cat# KK2802, Roche) and 1 μL of bisulfite-treated DNA (~15 ng/μl) for KSR- and FBS-iPSCs and TX 2i ESCs. PCR conditions and primer sequences were published previously[58] and can be found in Supplementary Table 2. Successfully amplified PCR products were purified and annealed with the sequencing primer for pyrosequencing using a PyroMark Q48 Autoprep instrument (Qiagen), according to the manufacturer's instructions. CpG methylation was quantified and analysed using Q48 Autoprep (v2.4.2 Build 3), PyroMark Autoprep Q48 Software (v4.2.1) and Firmware (v4.03) and OS version (v2.0.0).

## IMPLICON library preparation and analysis

IMPLICON was performed as previously described[44] in parental female and male F1 MEFs, respective KSR- and FBS-, KSR/FBS- and FBS+VitC-iPSCs, TX 2i ESCs, as well as day 12/24 female and male reprogramming intermediates/non-reprogramming intermediates. Briefly, following bisulfite conversion, a first PCR amplifies each region per sample in individual reactions, adding adapter sequences, as well as 8 random nucleotides (N₈) for subsequent data deduplication. PCR conditions and primers for this first step are listed in Supplementary Table 3. After pooling amplicons for each biological sample and clean-up using AMPure XP magnetic beads (Cat# A63880, Beckman Coulter), a second PCR completes a sequence-ready library with sample-barcodes for multiplexing. In this PCR reaction, barcoded Illumina adapters are attached to the pooled PCR samples ensuring that each sample pool receives a unique reverse barcoded adapter. Libraries were verified by running 1:30 dilutions on an Agilent bioanalyzer and then sequenced using the Illumina MiSeq platform to generate paired-end 250 bp reads using 10% PhIX spike-in as the libraries are of low complexity.

IMPLICON bioinformatics analysis was also performed as described[44], following the step-by-step guide of data processing analysis in [https://github.com/FelixKrueger/IMPLICON]. Briefly, data was processed using standard Illumina base-calling pipelines. As the first

step in the processing, the first 8 bp of Read 2 were removed and written into the readID of both reads as an in-line barcode, or Unique Molecular Identifier (UMI). This UMI was then later used during the deduplication step with "deduplicate bismark–barcode mapped_file.bam". Raw sequence reads were then trimmed to remove both poor quality calls and adapters using Trim Galore v0.5.0 (www.bioinformatics.babraham.ac.uk/projects/trim_galore/, Cutadapt version 1.15, parameters:–paired). Trimmed reads were aligned to the mouse reference genome in paired-end mode. Alignments were carried out with Bismark v0.20.0. CpG methylation calls were extracted from the mapping output using the Bismark methylation extractor. Deduplication was then carried out with deduplicate_bismark, using the–barcode option to take UMIs into account (see above). The data was aligned to a hybrid genome of BL6/CAST (the genome was prepared with the SNPsplit package - v0.3.4, [https://github.com/FelixKrueger/SNPsplit]). Following alignment and deduplication, reads were split allele-specifically with SNPsplit. Aligned read (.bam) files were imported into Seqmonk software v1.47 [http://www.bioinformatics.babraham.ac.uk/projects/seqmonk] for all downstream analysis. Probes were made for each CpG contained within the amplicon and quantified using the DNA methylation pipeline or total read count options. Downstream analysis was performed using Microsoft Excel spreadsheet software (v2206 Build 16. 0. 15330. 20144) and GraphPad Prism v8.0.1.

From the raw data deposited in GEO under the accession number GSE148067, the reads mapped to the following murine (mm10) genomic coordinates were excluded for consideration in this article for one of the following reasons: (1) regions that fail to reach the coverage threshold for the two parental alleles in a given sample (> 40 reads); (2) regions sequenced twice for which only the run with more reads was considered; (3) regions out of the scope of this article: for the samples *NNNN_4666*: Chr1:63261125-63264796, Chr2:174295708-174296349, Chr6:4746303-4746438, Chr15:72809673-72810197; for the samples *NNNN_4836*: Chr1:63261125-63261262; for the samples F FBS5 NPC, M FBS5 NPC, F FBS day12 T-/S + (1), F FBS day 12 T + /S- (1): Chr6:30737609-30737809; Chr10: 13091188-13091361; Chr11: 12025411-12025700; Chr17: 12742173-12742488; Chr18: 12972868-12973155; For F and M KSR-iPSC_5482: Chr11: 94970622-94970851. For *NNNN_5620*: Chr5: 144605801-144606063. Chr12: 39984798-39984915.

## RNA extraction
Total RNA was extracted from cells using NZYOL™ RNA Isolation Reagent (Cat# MB18501, NZYTech) and treated with DNase I (Cat# 04716728001, Roche) according to manufacturer's instructions.

## RT-qPCR
500 ng of DNase-treated RNA was reverse-transcribed into cDNA using random hexamers and a Reverse Transcriptase according to the manufacturer's protocol (Transcriptor High Fidelity cDNA Synthesis Kit, Cat# 05081963001, Roche). Quantitative real-time PCR was performed using iTaqTM SYBR® Green Supermix (Cat# 1725124, Bio-Rad) in an Applied Biosystems 7500 fast or ViiA 7 equipment to measure expression levels of several genes normalised to the *Gapdh* housekeeping gene. Primers and conditions are listed in Supplementary Table 2. The relative expression of each gene of interest was determined using the $2^{-\Delta\Delta ct}$ method.

## RT-PCR followed by Sanger sequencing
To analyse relative allelic expression of the *H19*, *Meg3* and *Snrpn* imprinted genes, cDNA was generated as described in RT-qPCR and amplified by PCR using the primers in Supplementary Table 2. The PCR product was cleaned using the NZYGelpure kit (Cat# MB01102, Nzytech) and sent for Sanger sequencing to STABVIDA sequencing company and data were visualised and analysed on a Chromas v2.6.2 software.

## RNA-seq
Quality of Dnase I-treated total RNA from biological triplicates of female MEFs, F KSR2, F KSR4, M KSR3, M KSR5 iPSCs and TX 2i ESCs was checked by 2100 Agilent Bioanalyser. Samples with RIN score above 9 were processed. RNA (1 µg) was used to generate strand specific polyA 250–300 bp insert cDNA libraries using the bespoke sequencing pipeline at Sanger Institute. Libraries were sequenced with Illumina HiSeq platform using single-end 50 bp mode.

RNA-seq raw FastQ data were trimmed with Trim Galore v0.6.1 (default parameters) and mapped to the mouse GRCm38 genome assembly using Hisat2 v2.1.0. Differentially expressed genes between female and male KSR-iPSCs were determined using both EdgeR v3.26.7[77] ($p$-value < 0.05 with multiple testing correction using Benjamimi and Hochberg correction) and intensity difference filter ($p$-value < 0.05 with multiple testing correction using Benjamimi and Hochberg correction), with the intersection between the two lists giving the high confidence differentially expressed genes. Clustering analysis in Fig. 1B used a Pearson correlation to calculate a distance matrix between all datasets which was then used to construct a neighbour joining tree.

Allele-specific alignments were performed by mapping to both CAST_EiJ and C57BL/6J (GRCm38) genomes, keeping reads that were specific for either genome and excluding those containing conflicting SNP information. Aligned read (bam) files were imported into Seqmonk software v1.47 [http://www.bioinformatics.babraham.ac.uk/projects/seqmonk] for all downstream analysis using standard parameters. Data was quantified at the mRNA level using strand-specific quantification of mRNA probes using the RNA-seq quantification pipeline in Seqmonk v1.47. For the heatmap in Fig. 3A showing allele-specific biases of imprinted genes, selection was based on the following criteria: (1) transcription from a single allele in MEFs (ratio: > 90%:10%), (2) Log2 RPKM > 1 expression in all iPSC replicates; (3) Normalised SNP-specific read counts > 5 in at least two of the three replicates of iPSCs (Supplementary Data 3).

## Statistics
Statistical analysis used for each experiment is indicated in the respective figure legend with $p$-values indicated or marked as * $p < 0.05$, ** $p < 0.01$ *** $p < 0.001$. The following tests were used: unpaired two-tailed Student's $t$-test (Fig. 2B and Supplementary Fig. 4E), two-way ANOVA followed by multiple comparisons corrected by the original FDR method of Benjamini and Hochberg (Fig. 5B, D and E), two-way ANOVA followed by Tukey's multiple comparisons test (Fig. 5C; Supplementary Fig. 6). Differential expression analysis of RNAseq data used both EdgeR and intensity difference filter ($p$-value < 0.05 with multiple testing correction using Benjamini and Hochberg correction for both) (Supplementary Fig. 1E and Supplementary Data 1).

## Reporting summary
Further information on research design is available in the Nature Research Reporting Summary linked to this article.

## Data availability
Source data are provided with this paper, and all the sequencing datasets produced in this study are available at Gene Expression Omnibus GSE148067 [https://www.ncbi.nlm.nih.gov/geo/query/acc.cgi?acc=GSE148067]. A reporting summary for this article is available as a Supplementary Information file. Source data are provided with this paper.

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

## Acknowledgements

We would like to thank Sérgio de Almeida, Miguel Casanova and Inês Milagre for critical reading of the manuscript, and the members of the S.T.d.R.'s team for helpful discussions. We also thank Tânia Carvalho and Pedro Ruivo for their help in histological analysis; Judith Webster at Babraham Institute for LC-MS measurements; Bethan Hussey at Sanger Sequencing and Kristina Tabbada at Babraham Institute for assistance with high-throughput sequencing; and the Bioimaging unit as well as Andreia Santos, Rute Gonçalves and Mariana Fernandes of the Flow Cytometry Facility of Instituto de Medicina Molecular João Lobo Antunes for their services and assistance. Work in S.T.d.R.'s team was supported by Fundação para a Ciência e Tecnologia (FCT) Ministério da Ciência, Tecnologia e Ensino Superior (MCTES), Portugal [IC&DT projects PTDC/BEX-BCM/2612/2014 and PTDC/BIA-MOL/29320/2017 as well as projects UIDB/04565/2020 and UIDP/04565/2020 of the Research Unit Institute from Bioengineering and Biosciences – iBB and LA/P/0140/2020 of the Associate Laboratory Institute for Health and Bioeconomy – i4HB]; S.T.d.R. and A.-V.G. are supported by assistant research contracts from FCT/MCTES (CEECIND/01234/2017 and CEECIND/02085/2018, respectively); M.A and A.C.R. are supported, respectively, by SFRH/BD/151251/2021 and SFRH/BD/137099/2018 PhD fellowships from FCT/MCTES. J.V.G.L is supported by COVID/BD/152624/2022 from FCT/MCTES. MAE-M was supported by a BBSRC Discovery Fellowship (BB/T009713/1) and is now supported by a Snow Medical Fellowship. F.K. is supported by the Babraham Institute Strategic Core Funding and A.M. by BBSRC BBS/E/B/000C0421. B.B.J. work was funded by Fundação para a Ciência e Tecnologia (FCT), and FEDER, LISBOA-01-0145-FEDER-028534, project co-funded by FEDER, through POR Lisboa 2020—Programa Operacional Regional de Lisboa. T.K. is supported by Janko Jamnik Doctoral Scholarship from National Institute of Chemistry.

## Author contributions

S.T.d.R. conceived the study, supervised the project and together with M.E.-M. secured funding. M.A. performed the characterization of all iPSCs and the majority of molecular biology experiments, generated KSR/FBS- and FBS + VitC-iPSCs and NPC lines, collected reprogramming intermediates by FACS, and prepared and analysed the IMPLICON experiments. J.v.G.L. generated MEFs and KSR-iPSCs, and T.K. generated and performed IMPLICON for the FBS-iPSCs and initial FACS experiments. A.C.R. helped with IMPLICON library preparation and molecular biology experiments. A.M. conducted IMPLICON experiments and DNA digestion for 5mC/5hmC measurements, as well as, M.E-M. who also analysed IMPLICON data. D.O. performed the 5mC/5hmC

measurements. S.M. maintained the i4F-BL6 mouse line and generated MEFs for KSR/FBS- and FBS+VitC-iPSCs. A-V.G. conducted the pyrosequencing experiments and helped with NPC generation. B.B.J. provided reprogramming expertise and access to the i4F-BL6 mouse lineage. F.K. and M.E.-M. conducted bioinformatic analysis of IMPLICON and RNAseq. M.A and S.T.d.R. wrote the manuscript with contributions of all authors.

## Competing interests

A.M. and F.K. are Altos Labs employees. The other authors declare no competing interests.

## Additional information

[1]iBB - Institute for Bioengineering and Biosciences and Department of Bioengineering, Instituto Superior Técnico, Universidade de Lisboa, Lisbon, Portugal. [2]Associate Laboratory i4HB Institute for Health and Bioeconomy, Instituto Superior Técnico, Universidade de Lisboa, Lisbon, Portugal. [3]Instituto de Medicina Molecular, João Lobo Antunes, Faculdade de Medicina, Universidade de Lisboa, Lisboa, Portugal. [4]Epigenetics Programme, Babraham Institute, Cambridge CB22 3AT, United Kingdom. [5]Peter MacCallum Cancer Centre, Melbourne, Victoria 3000, Australia. [6]Sir Peter MacCallum Department of Oncology, The University of Melbourne, Victoria 3010, Australia. [7]Department of Anatomy and Physiology, The University of Melbourne, Victoria 3010, Australia. [8]National Institute of Chemistry, Ljubljana, Slovenia. [9]NOVA Medical School|Faculdade de Ciências Médicas, NMS|FCM, Universidade Nova de Lisboa, Lisboa, Portugal. [10]Bioinformatics Group, Babraham Institute, Cambridge CB22 3AT, United Kingdom. [11]Altos Labs, Cambridge, United Kingdom. [12]Mass Spectrometry Facility, The Babraham Institute, Cambridge, United Kingdom. [13]Genetics and Developmental Biology Unit, Institut Curie, INSERM U934, CNRS UMR3215, PSL University, Paris, France. [14]Department of Medical Sciences and Institute of Biomedicine - iBiMED, University of Aveiro, 3810-193 Aveiro, Portugal. ✉e-mail: simao.rocha@tecnico.ulisboa.pt

