## [Peer Review File · Nature Communications]

Imprinting fidelity in mouse iPSCs depends on sex of donor cell
and medium formulationREVIEWER COMMENTS

Reviewer #1 (Remarks to the Author):

In their manuscript entitled “Sex of donor cell and reprogramming conditions predict the extent and nature of imprinting defects in mouse iPSCs” Arez, da Rocha use a transgenic murine reprogramming system to address the epigenetic stability of a subset of imprinted gene loci in different sets of pluripotent cells. Imprinted defects are major roadblocks for the use of iPSCs as research tools and in the clinic, but their origin remain incompletely understood. To address this important issue, the authors compare the effects of two common culture media (KSR vs FBS) and of sex (male vs female) of the fibroblasts used for iPSC derivation on imprinting. Based on DNA methylation analysis and gene expression profiling they conclude that specific imprinting errors are associated with both parameters under investigation. Specifically, they report that a) female sex is associated with widespread DNA hypomethylation at imprinted loci in either condition and b) in male iPSCs, KSR results in DNA hypomethylation at multiple imprinted genes while FBS results in hypermethylation at only few loci. The authors’ findings touch upon several interesting aspects and raise a cautionary note regarding the use of KSR during iPSC derivation, which is a relevant finding considering the widespread use of the media supplement. Nevertheless, in my opinion their study currently lacks the necessary degree of novelty to recommend publication in Nature Communications. This is in large part due to the fact that some of their major conclusions are not satisfyingly supported by their experimental approaches. Addressing the points listed below would substantially strengthen the manuscript.

1. The experimental approach taken for iPSC derivation (Figure 1A and 4A) does not allow distinction between changes in DNA methylation triggered during reprogramming (Days 0 to 12) and changes introduced during subsequent culture (Days 12 to 50, the analysis time point). This devalues the authors’ claim that reprogramming conditions affect imprinting status in derivative iPSCs. To support these claims, iPSCs would have to be generated and compared that experienced different reprogramming conditions (FBS vs KSR) both identical conditions of subsequent maintenance culture. In addition, isogenic ESCs (i.e. pluripotent cells that never underwent reprogramming) derived and expanded under the two conditions could be analyzed.

2. Similarly, it is unclear whether sex actually affects imprinted loci during reprogramming or thereafter. The vulnerability of female iPSCs/ESCs for ICR hypomethylation during maintenance culture is well established.

3. While the analysis of DNA methylation at imprinted genes in the pluripotent state is convincing, it is unclear how many of the observed perturbations indeed reflect irreversible imprinting defects (as the authors state and imply). The authors should test whether both hypermethylation and hypomethylation defects at all loci are maintained upon differentiation.

4. The authors claim that KSR formulation leads to more pronounced imprinting defects than FBS media. In addition to the unclear reversibility of these defects (see point 3), their significance with respect to developmental potential of affected iPSCs are unclear. The authors should address this by conducting functional assays such as blastocyst injection experiments.

5. It would be desirable if the authors provided either mechanistic insights into their findings (for example by identifying a relevant regulator responsible DNA hypomethylation downstream of KSR) or develop a reprogramming/culture approach that circumvents or at least substantially reduces imprinting defects in iPSCs.

Additional comments

1. It is unclear why 2i ESCs were used as controls for this study.

2. Were all male/female iPSCs derived from the same MEF prep?

3. For how many generations have i4F-BL6 been backcrossed to BJ6? The published effect of (B6J) background on imprint stability should be discussed (PMID 32187532).
4. The resolution of LC-MS (Fig. 6A,C) is not sufficient to rule out widespread alterations at non-imprinted loci. The authors' analysis of retrotransposons supports the notion of additional genomic regions being affected (Fig. 6B)
5. The authors state on page 9 that at least one of their iPSCs clones showed major chromosomal abnormalities. Considering the observed variability between iPSC clones, chromosome counts should be conducted in all clones and lines with major abnormalities of autosomes excluded. In addition, the author should clarify whether their female lines have lost one of the two active X chromosomes at the time point of analysis.
6. It would be helpful to present efficiencies of iPSC derivation in both KSR and FBS conditions.

Reviewer #2 (Remarks to the Author):

Arez et al presents their work on imprinting defects in mouse iPSC (fibroblast-derived) highlighting sex of donor cell and culture conditions to be associated with these defects. Since the introduction of the iPSC protocol decades ago there have been extensive work showing not only heterogeneity in pluripotency but also imprinting aberrations with key loci such as Dio3 and Zrsr1 being clearly impacted in iPSCs. However, the authors state that no systematic assessment has been made taken sex into account. Although the sex aspect is interesting and novel, I find the paper still very limited and would not consider the presented report as being comprehensive enough to be considered a systematic analysis.

My main concerns are as follows:

1. Only 8/25 loci are studied (with no rationale for the inclusion of only eight) of which many already have been shown to be impacted in iPSC and with current methodologies as well as the relatively low number of samples (Total N~25) the complete "catalog" should have been included.
2. The study is limited to mouse iPSC only despite many others have shown similar (general) phenomena in human iPSCs. The female-specific effect is not known to occur in human iPSCs.
3. The observed sex-specific effects in imprinting defects is not linked to differences in pluripotency or differentiation despite the observations of clear inter-individual differences in methylation aberrations within each group.
4. No treatment strategy or optimized culture conditions are presented rescuing the observed imprinting defects (as shown for e.g. the DLK1-Dio3 loci)

My additional minor concerns are as follows:

1. The authors show global changes in gene expression in female vs male iPSC:
 - a. They report that females have in general more genes upregulated than males and the authors are not reporting any discussions or interpretations why this is observed. Did they observe the same pattern for the FBS-iPSC?
 - b. Stable 2 is incomplete as no pvalues or fold-changes are reported

Reviewer #3 (Remarks to the Author):

In this manuscript, Arez et al. present a thorough analysis of imprinting integrity in mouse iPSCs by taking advantage of IMPLICON method, which gives the ability to create a base-resolution methylation map on targeted regions. The authors initially generated iPSCs from MEFs which were derived from a cross between two genetically distant mouse strains. This enabled them to

discriminate between the two parental alleles when using IMPLICON and measure the allele-specific methylation levels along 8 imprinted clusters. The authors then generated iPSCs from both female and male MEFs, and under different culture conditions compared the imprinting status in each locus. They also obtained female mouse ESCs cultured under 2i, which is known to lead to global demethylation, including in imprinted loci. These comparisons revealed a few main findings. First, the authors were able to show that when compared with their parental MEFs, iPSCs were hypomethylated in all the imprinted regions which they analyzed. They also noticed a strong bias towards a hypomethylated state in female cells, which was closer to the methylation levels observed in 2i ESCs. Next, they compared between iPSCs that were obtained and grown in KSR vs. FBS media and showed that FBS causes reduced levels of methylation-loss. This was especially evident in male iPSCs which showed methylation levels which were very close to the ones observed in their parental MEFs. Notably, the IMPLICON resolution allowed demonstrating that methylation loss was caused by heterogeneity in the culture. Importantly, the authors showed that the methylation loss in imprinted control regions resulted in the biallelic expression of imprinted genes. Finally, they showed that the differences between the sexes and media were also evident when examining global methylation levels, albeit to a lesser extent, highlighting the sensitivity of the imprinting regions for the imprint loss. The authors also looked on methylation levels along repetitive elements, which showed variable methylation states with some exhibiting similar and some distinct methylation patterns than the ones observed in the imprinted clusters.

This article uses a well-designed methodology to give a high-resolution map of imprinting aberrations in iPSCs. It also highlights the more aggressive imprinting aberrations that are caused by the KSR compared to the serum-containing (FBS) media. However, it should be noted that many of the key findings which are highlighted in this manuscript are not novel and are already well-known. Specifically, it was previously reported that female cells go through a more aggressive methylation loss in imprinted loci both in mouse ESC (Choi et al., 2017; Zvetkova et al., 2005, both referenced in the paper) and iPSC (Pasque et al., 2018, referenced in the paper). It was also reported that while male mouse ESCs grown on serum-containing media are relatively immune to imprinting aberrations, 2i ESCs (both male and female) together with female ESCs grown on serum show a global reduction of methylation that includes imprinted loci (Choi et al., 2017).

Thus, some major concerns should be addressed for this manuscript to be considered for publishing in a high-impact journal:

1. The authors point to several genes which are part of the DNA methylation machinery and are differentially expressed between iPSCs in FBS vs. KSR media, as possible drivers for the differences observed between these conditions. However, these results are only correlative. Thus, genetic perturbation or at least chemical inhibition of the suspected genes should be performed to see whether these perturbations extinguish the different methylation patterns in KSR compared with FBS. Specifically, the authors highlight the differences in the expression of Tet1/2 enzymes which were upregulated in the KSR iPSCs. They suggest that either the upregulation of Tet1/2 or their higher activity which is triggered by the presence of ascorbic acid that promotes their activity could lead to the lower methylation levels observed in the KSR media. Thus, aside for perturbing Tet1/2, the addition of ascorbic acid to the FBS media should be performed in order to see if it causes a more aggressive methylation loss.
2. As the sensitivity of female iPSCs and ESCs to methylation loss is already well-established, the authors should include in the analysis ESCs which are not only grown on 2i, but also on KSR and FBS. This will demonstrate whether the reprogramming of MEF to iPSCs results in higher aberration levels than what is observed in ESCs under FBS or KSR conditions. In this context, following the methylation dynamics in different time-points along the reprogramming process in every condition could strengthen the results. Following the cells along the reprogramming process will also enable to identify whether the differences in the imprinting aberrations between KSR and FBS arise at the same time-point during reprogramming or following reprogramming.
3. The authors also point to X-linked genes which were differentially expressed between female and

male iPSCs as possible drivers for the methylation difference between them. However, these results are also correlative and should be established by perturbation of the genes.

4. In Figure S3A, the percentage of reads obtained from each chromosome in each cell-line is presented. However, this presentation does not give the ability to fully assess whether the cells have a normal karyotype. In this context, it is not clear why the male iPSCs do not have any reads obtained from the Y chromosome. Thus, a karyotype for each line should be provided. This will also enable to understand whether imprinting aberrations are connected to chromosomal instability.

5. In Figure S5B, the expression of Meg3 for every iPSC line is shown in order to establish that lines which showed more aggressive methylation erasure in their imprinting control regions also express higher levels of imprinted genes. However, this should be performed for other genes and not only for Meg3. Furthermore, the correlation between the methylation in an imprinting control region to the expression levels of genes under the control of the region should be plotted as a correlation plot which is much easier to interpret.

6. In this manuscript the focus was on 8 imprinted clusters while the original IMPLICON paper included 9 clusters (Klobučar et al., 2020, referenced in the manuscript). The authors should therefore explain why they did not include the last region that was included in the first paper.

Minor comment:

1. Figures S6A, S6B should have separate legends.

POINT-BY-POINT RESPONSE

Dear Reviewers,

Thank you for the opportunity to revise our manuscript on imprinting defects in mouse iPSCs with the tracking number NCOMMS-20-44104. We were very pleased that Reviewers considered that “*This article uses a well-designed methodology to give a high-resolution map of imprinting aberrations in iPSCs*” (reviewer 3), that our “*...findings touch upon several interesting aspects and raise a cautionary note regarding the use of KSR during iPSC derivation...*” (reviewer 1), and that “*the sex aspect is interesting and novel*” (reviewer 2).

We deeply thank the Reviewers for the careful analysis of our work and for the constructive comments that genuinely helped to guide us through a long revision process. We believe we have substantially improved our manuscript, which now includes 6 new experiments, 2 new Result sections, 2 new main Figures, 3 additional Suppl. Figures plus 5 new panels added to the original Suppl. figures, as well as new IMPLICON datasets uploaded on GEO.

The main revisions include:

- Inspection of imprinting fidelity during iPSC reprogramming of female and male iPSCs to determine the timing of imprinting errors (Fig. 6A-E; Fig. S7C-G).
- FBS-to-KSR and KSR-to-FBS medium swaps to determine the impact of medium formulations on imprinting maintenance (Fig. S7A-B).
- Differentiation of iPSCs into Neural Precursor Cells (NPCs) to assess persistence of imprinting defects in iPSC differentiated derivatives (Fig. S8).
- Karyotyping to analyse chromosomal stability of reprogrammed cells (Fig. S1C; Fig. S4C).
- Evaluation of 5 additional imprinted loci for a more thorough analysis of imprinting fidelity in iPSCs (Fig. 2B; Fig. S4E; Fig. 5A; Table S3).
- Reprogramming in KSR/FBS 1:1 and FBS+VitC conditions as potential strategies to mitigate/correct imprinting defects in iPSCs (Fig. 7A-C; Fig. S9A-B).

Text changes are highlighted in red in the revised manuscript. Our Point-by-point response to the Reviewers is found below (our replies are marked in blue).

We hope you consider that our revised manuscript has considerably improved and that the new data greatly strengthen our findings on imprinting defects of iPSCs.

Looking forward to hearing from you.

Best wishes,

Simão Rocha, on behalf of all authors

Reviewer #1 (Remarks to the Author):

In their manuscript entitled “Sex of donor cell and reprogramming conditions predict

the extent and nature of imprinting defects in mouse iPSCs” Arez, da Rocha use a transgenic murine reprogramming system to address the epigenetic stability of a subset of imprinted gene loci in different sets of pluripotent cells. Imprinted defects are major roadblocks for the use of iPSCs as research tools and in the clinic, but their origin remain incompletely understood. To address this important issue, the authors compare the effects of two common culture media (KSR vs FBS) and of sex (male vs female) of the fibroblasts used for iPSC derivation on imprinting. Based on DNA methylation analysis and gene expression profiling they conclude that specific imprinting errors are associated with both parameters under investigation. Specifically, they report that a) female sex is associated with widespread DNA hypomethylation at imprinted loci in either condition and b) in male iPSCs, KSR results in DNA hypomethylation at multiple imprinted genes while FBS results in hypermethylation at only few loci. The authors’ findings touch upon several interesting aspects and raise a cautionary note regarding the use of KSR during iPSC derivation, which is a relevant finding considering the widespread use of the media supplement. Nevertheless, in my opinion their study currently lacks the necessary degree of novelty to recommend publication in Nature Communications. This is in large part due to the fact that some of their major conclusions are not satisfyingly supported by their experimental approaches. Addressing the points listed below would substantially strengthen the manuscript.

1. The experimental approach taken for iPSC derivation (Figure 1A and 4A) does not allow distinction between changes in DNA methylation triggered during reprogramming (Days 0 to 12) and changes introduced during subsequent culture (Days 12 to 50, the analysis time point). This devalues the authors’ claim that reprogramming conditions affect imprinting status in derivative iPSCs. To support these claims, iPSCs would have to be generated and compared that experienced different reprogramming conditions (FBS vs KSR) both identical conditions of subsequent maintenance culture. In addition, isogenic ESCs (i.e. pluripotent cells that never underwent reprogramming) derived and expanded under the two conditions could be analyzed.

We thank the reviewer for raising these important points that we have now addressed in the revised version. We assessed the imprinting status at day 12 and day 24 of reprogramming for both male and female cells generated under the FBS protocol (Fig. 6A). We performed IMPLICON for 6 imprinted loci (Fig. 6C), one methylated and one unmethylated control region (Fig. S7F; Table S3), relative allelic expression levels using RT-PCR followed by Sanger sequencing of the *H19* and *Snrpn* imprinted genes (Fig. 6D; Fig. S7G) and we measured global levels of 5mC and 5hmC by LC-MS (Fig. S7E).

Concerning female iPSCs, hypomethylation imprinting errors were not apparent at day 12, with first signs of errors seen at day 24 that become clearer by day 50 (Fig. 6C). Accordingly, reactivation of the silenced allele of both *H19* and *Snrpn* is also first seen on day 24 and is only fully evident by day 50 (Fig. 6D). Therefore, hypomethylation defects in female iPSCs are uncoupled from reprogramming. Importantly, hypomethylation defects at imprinting regions did not follow the global DNA methylation dynamics during reprogramming. A decrease of 5mC is readily seen at day 12, when a peak of 5hmC and the first signs of X-chromosome reactivation are seen (Fig. 6B; Fig. S7E). To corroborate these results, the *Prickle1*

methyated control region in our IMPLICON method closely follows this global DNA methylation dynamics (Fig. S7F). The fact that imprinting defects do not follow global DNA methylation dynamics and initially resist demethylation is a novel and unexpected finding that we now highlight in the revised manuscript. Complete loss of imprinting is only seen in female “fully reprogrammed iPSCs” at day 50 (Fig. 6C), when *XIST* is no longer expressed (Fig. 6B), a hallmark of pluripotency. In conclusion, our results show that imprinted regions are initially resistant to the first demethylation wave during reprogramming, but ultimately succumb to loss of imprinting due to persistent low 5mC levels, characteristic of female murine iPSCs/ESCs (Fig. 6C; Fig. S7E) (Zvetkova et al., 2005 -PMID: 16244654; Pasque et al., 2018 -PMID: 29681539).

Concerning male iPSCs, hypermethylation errors start to be evident in the *Dlk1-Dio3* locus at day 12 (Fig. 6C). These results match previous results using a similar reprogramming system (Stadtfield et al, 2012). However, thanks to the single-molecule resolution of IMPLICON, we gain more insights into the hypermethylation process. We observed that at day 12, in the majority of cells, the maternal ICR is hemi-methylated containing both unmethylated and methylated CpGs, that only at day 24 will resolve into fully methylated (and to a lesser extent fully unmethylated) state (Fig. 6E). Hypermethylation of *Dlk1-Dio3* at day 12 is also seen in female reprogramming cells as well as in both male and female non-reprogramming intermediates suggesting that it might be linked to the activation of the Yamanaka cassette. These new findings are documented in the text and depicted in Fig. 6C and reveal that hypermethylation errors at the *Dlk1-Dio3* locus are induced by reprogramming. For the *Igf2-H19* region, the mild hypermethylation errors seem to have a slightly slower kinetics, but are fully established by day 24. Again, and in contrast to the *Prickle1* gene, these hypermethylation imprinting errors do not follow the general dynamics of DNA methylation during male reprogramming (Fig. 6C; Fig. S7E-F). These imprinting errors seem to be reprogramming-induced and explain why they have been previously noticed to be more frequent in mouse iPSCs than ESCs (Stadtfield et al., 2010 - PMID: 20418860; Yagi et al., 2019 - PMID: 31056481).

Unfortunately, we do not have the expertise in the lab to generate isogenic hybrid ESCs. Nevertheless, we wanted to address the valid point of the reviewer concerning the impact of medium conditions in imprinting stability during culture maintenance of iPSCs. For that, we decided to swap medium conditions and culture our FBS-iPSCs in KSR medium and KSR-iPSCs in FBS medium for up to 10 passages (around 20 days) and analyse by IMPLICON 6 imprinted loci, one methylated and one unmethylated control region (Fig. S7A-B; Table S3), and by relative allelic expression levels using RT-PCR followed by Sanger sequencing of the imprinted genes, *H19* and *Snrpn* (Fig. S7A-B). This experiment was actually very informative. We observed that culturing FBS-iPSCs in KSR medium resulted in imprinting instability that ranges from milder to stronger effects (Fig. S7A). This was also the reason why we did not investigate imprinting defects during KSR derivation of iPSCs, since the KSR medium by itself induces imprinting errors. We observed the opposite when we swapped the medium to FBS, as we did not see differences in methylation or relative allelic expression of imprinted genes in the time frame of this experiment (Fig. S7B).

2. Similarly, it is unclear whether sex actually affects imprinted loci during

reprogramming or thereafter. The vulnerability of female iPSCs/ESCs for ICR hypomethylation during maintenance culture is well established.

This is also an important aspect raised by the reviewer. We have now addressed this issue by performing IMPLICON, relative allele-specific expression analysis of imprinted genes and 5mC/5hmC measurements as specified in point 1. Our data show that hypomethylation defects in female iPSCs are indeed uncoupled from reprogramming. Importantly, this analysis led to a novel finding, which was that loss of methylation at imprinted loci is not a direct consequence of female iPSC hypomethylation, but is rather passively lost with time in culture (Fig. 6C; see point 1).

3. While the analysis of DNA methylation at imprinted genes in the pluripotent state is convincing, it is unclear how many of the observed perturbations indeed reflect irreversible imprinting defects (as the authors state and imply). The authors should test whether both hypermethylation and hypomethylation defects at all loci are maintained upon differentiation.

We thank the reviewer for this important point. In order to understand whether imprinting defects were irreversible upon differentiation, we differentiated iPSCs with hypomethylation and hypermethylation defects (F FBS5 and M FBS5, respectively) into Neural Progenitor Cells (NPCs) (Fig. S8A) and analysed them by IMPLICON and relative allelic expression levels using RT-PCR followed by Sanger sequencing of *H19* and *Snrpn* genes (Fig. S8B-C). As our IMPLICON results clearly demonstrate, both hypo- and hypermethylation defects were maintained in NPCs, while the levels of methylation of a non-imprinted gene (*Prickle1*) recovered to the levels seen in MEFs (Fig. S8B). We also confirmed persistence of biallelic expression of imprinted genes in female NPCs and absence of *Meg3* expression in male NPCs, as previously observed in female and male iPSCs, respectively (Fig. 4C; Fig. S5A; Fig. S8C). Therefore, our data shows the irreversibility of imprinting defects in mouse iPSCs, which was also previously documented for human iPSCs by other authors (Bar et al., 2017 - PMID: 28467909; Nazor et al., 2012 - PMID: 22560082).

4. The authors claim that KSR formulation leads to more pronounced imprinting defects than FBS media. In addition to the unclear reversibility of these defects (see point 3), their significance with respect to developmental potential of affected iPSCs are unclear. The authors should address this by conducting functional assays such as blastocyst injection experiments.

We have addressed developmental potential by performing teratoma assays. Our analysis showed well-differentiated teratomas in KSR- and FBS-iPSCs suggesting that *“imprinting dysregulation in iPSCs does not impede the major axes of lineage commitment”* (in the Discussion). We agree it was important to show that imprinting defects remained in differentiated derivatives of our iPSCs (point 3). However, addressing developmental potential of affected iPSCs by blastocyst injections is not a trivial experiment and we don't have the expertise to do it. Also, our FBS- and KSR-iPSCs have several imprinting errors with some variability which will make the interpretation of such experiments extremely difficult. Further functional assays are

out of the scope of this manuscript and would deserve another in depth analysis, which will take quite some time to be accomplished. We rather concentrate on exploring ways to correct/mitigate these imprinting errors with alternative reprogramming approaches (point 5).

It is important to point out that many of the imprinting defects observed in our iPSCs and maintained in NPCs, are known to lead to mouse imprinting phenotypes and imprinted diseases. This has now been stated in the Discussion: “*These imprinting defects are known to cause developmental/metabolic phenotypes in mice and cause imprinted disorders in humans (Tucci et al., 2019; Monk et al., 2019)*”. We have also acknowledged in the Discussion the importance of doing blastocyst experiment injections to study the developmental potential of our generated iPSCs: “*Generation of chimeric mice or all-iPSC mice through blastocyst injections would provide a more refined way to address the developmental potential of distinct imprinting defects that result from the reprogramming of female and male iPSCs in different conditions and should be explored in the future in a controlled manner.*”

5. It would be desirable if the authors provided either mechanistic insights into their findings (for example by identifying a relevant regulator responsible DNA hypomethylation downstream of KSR) or develop a reprogramming/culture approach that circumvents or at least substantially reduces imprinting defects in iPSCs.

Our study focuses on describing the effect of reprogramming/medium conditions in imprinting stability in female and male iPSCs. Therefore, we decided to develop alternative reprogramming approaches to correct or minimise imprinting errors. It was beyond our scope to provide mechanistic insights into the specific regulators of the multiple defects that we observed.

We concentrated our efforts on male iPSCs, as our data (point 1 and 2) shows that hypomethylation errors in female cells are not reprogramming-induced. Based on the knowledge about the opposite effects of KSR and FBS on imprinting in male iPSCs, we decided to test whether a 1:1 ratio of KSR/FBS or FBS with vitamin C (FBS+VitC), a component of the KSR medium that activates TET demethylase enzymes, would correct or at least mitigate the hypermethylation imprinting defects in male FBS-iPSCs. The 1:1 KSR/FBS formulation was able to fully recover the hypermethylation phenotype at *Igf2-H19* and, to a certain extent, at *Dlk1-Dio3* loci. This success was offset by the hypomethylation defects on the methylated allele in the majority of loci we studied (Fig. 7A). The FBS+VitC formulation did not induce hypomethylation defects (except for a mild effect in *Mcts2-H13*) and was able to moderately mitigate the hypermethylation defect at the *Dlk1-Dio3* locus (Fig. 7A). Close inspection of the single-molecule IMPLICON datasets showed that 3 out of 6 FBS+VitC-iPSCs have a mixed population where some cells exhibit normal imprinting pattern, while others exhibit a hypermethylated phenotype at the *Dlk1-Dio3* locus (Fig. 7B). Using serial dilutions, we could isolate at least one clone (clone 5) presenting normal imprinting judged by the expression of the *Meg3* gene (Fig. 7C). Curiously, this same clone also showed *H19* expression levels comparable to M FBS2 with normal imprinting at *Igf2-H19* locus (Fig. S9B). In conclusion, these data show that FBS+VitC formulation mitigates imprinting errors in male iPSCs. The concentration of VitC that we used

was comparable to the one used by Stadtfeld et al 2012 (PMID: 22387999). In their system, VitC led to a better improvement at the *Dlk1-Dio3* locus, but they have not used a sensible method as IMPLICON and have not checked VitC effect on other imprinted regions. In the discussion, we debate about these differences and suggest that playing with different VitC concentrations and other parameters could help to further improve the generation of imprinting error-free iPSCs: “*Differences in the genetic background or other technical aspects (e.g., different FBS batches) might have contributed to these differences. Playing with higher concentrations of VitC is an avenue to explore in the future, bearing in mind that any improvement in hypermethylation defects at the Dlk1-Dio3 or Igf2-H19 could be offset by hypomethylation defects in other imprinted loci. In conjunction to VitC adjustments, alternative reprogramming setups could also be considered such as, the use of different stoichiometry of the 4 Yamanaka factors or even the exclusion of the Oct4 from the Yamanaka cocktail which seem to have a positive impact on imprinting maintenance (Carey et al., 2011; Velychko et al., 2019)*”.

Additional comments

1. It is unclear why 2i ESCs were used as controls for this study.

2i ESCs act as technical control for cells with full erasure of imprinting as opposed to parental MEFs with normal imprinting. We have now made this clearer in the Results section on “Female and male KSR-iPSCs show loss of methylation at imprinted regions” by adding the following: “... *and in the TX 2i ESC line expected to have erased methylation imprints due 2i-induced demethylation (Ficz et al., 2013; Habibi et al., 2013; Leitch et al., 2013; von Meyenn et al., 2016)*”.

2. Were all male/female iPSCs derived from the same MEF prep?

Yes, the male and female KSR- and FBS-iPSCs came from the same male or female MEF preparation. For the novel experiments, KSR/FBS-iPSCs and FBS+VitC-iPSCs, we needed to recur to new MEF preparations, as we did not have extra biological material from the original MEF preps. This information is now clarified in the Materials and Methods section “Reprogramming of MEFs”: “*The same batch of female and male MEFs derived from the single embryos of the same progeny were used to generate KSR-iPSCs and FBS-iPSCs. Another batch of male hybrid MEFs from another progeny were used to generate KSR/FBS-iPSCs and FBS+VitC-iPSCs.*”

3. For how many generations have i4F-BL6 been backcrossed to BJ6? The published effect of (B6J) background on imprint stability should be discussed (PMID 32187532).

i4F-BL6 has been generated (Abad et al 2013; PMID: 24025773) and maintained on the BL6 background for more than 6 years in our lab. We showed that BL6 background does not have an effect on imprint stability in both female and male MEFs on this background (Fig. 2B; Fig. S4E). This information is now added in the Materials and Methods section on “Mice strains”: “*The i4F-BL6 mouse colony was always maintained on the BL6 background during this study, except when crossed*

with CAST animals to obtain F1 embryos (see Generation and maintenance of F1 hybrid MEFs)."

We also discuss the susceptibility of the *Dlk1-Dio3* locus to hypermethylation defects in pluripotent stem cells in a B6 background (Swaney et al., 2020; PMID: 32187532) and how we think that our reprogramming-induced defects in the *Dlk1-Dio3* are not simply explained by the hybrid genetic background we use as this was seen before by others in a different genetic background (Yagi et al., 2019; PMID: 31056481). We added the following to the discussion: *"Male FBS-iPSCs have mild and strong hypermethylation defects in the paternally methylated Igf2-H19 and Dlk1-Dio3 loci, respectively. Recently, it has been documented that the Dlk1-Dio3 locus is more sensitive to hypermethylation defects on pluripotent stem cells of the BL6 mouse strain (Swaney et al., 2020), which is the genetic background of the maternal locus in our hybrid BL6/CAST iPSCs. Our results match the ones observed by other authors using a different hybrid cross, 129X1/SvJ x MSM/Ms, for both Igf2-H19 and Dlk1-Dio3 loci (Yagi et al., 2019), showing that these intergenic paternally methylated ICRs are particularly susceptible to reprogramming-induced errors in different genetic backgrounds."*

4. The resolution of LC-MS (Fig. 6A,C) is not sufficient to rule out widespread alterations at non-imprinted loci. The authors' analysis of retrotransposons supports the notion of additional genomic regions being affected (Fig. 6B)

Our paper is focused on imprinting, but we cannot rule out the fact that other non-imprinted regions are also affected. Examples are the retrotransposon element L1-T that lost methylation potentiated by KSR conditions, L1-A and IAP as well as the *Prickle1* gene (a non-imprinted region used as methylated control) that drop its methylation levels to a certain extent in female iPSCs. This information is described in the paper on the section renamed "Biological sex and medium formulation impacts global 5mC/5hmC levels": *"In summary, while a modest decrease in the level of 5mC and in methylation of IAPs and L1-A elements could be discerned for female iPSCs (Fig. 5B-C), a strong effect of KSR medium in demethylating L1-T retrotransposons was detected (Fig. 5C). In conclusion, with the exception of L1-T retrotransposons, ICRs are among the most affected loci upon reprogramming with more pronounced effects on DNA methylation than the ones observed at repetitive elements"*.

Our data on NPCs show the persistence of imprinting defects upon differentiation, while the non-imprinted *Prickle1* gene shows full rescue of methylation to MEF levels (Fig. S8B). Therefore, while several genes can be affected by the DNA methylation dynamics through reprogramming/maintenance of the stem cells, imprints are notorious for not being rescued (repeats were not tested but they are also known to be prone to fluctuations in DNA methylation throughout development). We have reiterated this in the results section "Reprogramming strategies to generate male iPSCs without imprinting defects": *"IMPLICON revealed the persistence of both hypo- and hypermethylation errors and abnormal expression of imprinted genes in the corresponding NPC line (Fig. S8B-C). In contrast, the Prickle1 non-imprinted gene recovered the methylation levels seen in MEFs. While different genomic regions adjust their methylation levels in response to environmental cues (reprogramming/stem cell maintenance/differentiation), abnormal changes in the*

epigenetic status of imprinted genes persevere, which can have long-lasting consequences as previously shown (Bar et al., 2017; Nazor et al., 2012)."

5. The authors state on page 9 that at least one of their iPSCs clones showed major chromosomal abnormalities. Considering the observed variability between iPSC clones, chromosome counts should be conducted in all clones and lines with major abnormalities of autosomes excluded. In addition, the author should clarify whether their female lines have lost one of the two active X chromosomes at the time point of analysis.

The reviewer raised a very important point. Indeed, chromosomal abnormalities could arise and we specifically found one occasion where that happened (F KSR2 iPSC). In this revised version, we have performed karyotyping for the 8 most relevant female and male KSR- and FBS-iPSCs. Our data show that no major chromosomal abnormalities are seen in our iPSCs (Fig. S1C; Fig. S4C). Curiously, F KSR2 iPSC had a normal karyotype (Fig. S1C) and allele-specific RNA-seq analysis show that the loss of reads on the paternal chr6 was compensated by the duplication of the number of reads on the maternal chr6, suggesting a case of maternal uniparental disomy of chr6 (Fig. S3B). This explains the exclusive maternal expression of *Sgce*, *Peg10* and *Mest* genes. We also saw that the F FBS1 iPSC line became XO: presented a karyotype of 39 chromosomes (Fig. S4C), express half of the levels of X-linked genes compared to other female iPSC lines (Fig. S4D) and show only one X-chromosome by X-chromosome painting - Fig. 1 for reviewers, below). Loss of an X chromosome is commonly observed in female ESCs/iPSCs (Zvetkova et al., 2005 -PMID: 16244654; Pasque et al., 2018 -PMID: 29681539). In this XO female iPSC line, we saw that hypomethylation imprinting defects persisted, although they were milder (Fig. 4B; Fig. 5A, Table S3). This suggests that loss of the X chromosome in this line might have been a late event (between day 24-50) as hypomethylation defects are already present, but did not reach its maximum (Fig. 6C).

In a nutshell, our iPSCs do not present major chromosomal abnormalities and do not explain the imprinting defects which have very clear trends dictated by sex of donor cell and medium formulation.

6. It would be helpful to present efficiencies of iPSC derivation in both KSR and FBS conditions.

We did not see remarkable differences in the efficiency of iPSC derivation in the different conditions used in our study and obtained reprogramming efficiencies similar to what has been previously documented (Bernardes de Jesus et al., 2018). The different experiments, KSR, FBS, 1:1 KSR/FBS and FBS+VitC derivations were not done at the same time and so we are unable to perform a robust and systematic comparison in efficiencies between the different conditions in this study.

Reviewer #2 (Remarks to the Author):

Arez et al presents their work on imprinting defects in mouse iPSC (fibroblast-derived) highlighting sex of donor cell and culture conditions to be associated with

these defects. Since the introduction of the iPSC protocol decades ago there have been extensive work showing not only heterogeneity in pluripotency but also imprinting aberrations with key loci such as Dio3 and Zrsr1 being clearly impacted in iPSCs. However, the authors state that no systematic assessment has been made taken sex into account. Although the sex aspect is interesting and novel, I find the paper still very limited and would not consider the presented report as being comprehensive enough to be considered a systematic analysis.

We thank the reviewer for raising this point. We would like to point out that no previous study has looked at imprinting stability at multiple loci in both female and male isogenic iPSCs under 2 independent reprogramming conditions (see Table S1). In the revised version, we have now extended our analysis from 8 to 13 imprinted loci using IMPLICON and we also reprogrammed male cells using two additional reprogramming conditions (KSR/FBS and FBS+VitC). To our knowledge, this is the most comprehensive study of imprinting defects in mouse iPSCs. In any case, we understand from the author that “*systematic analysis*” might not be the accurate way to qualify our study. Therefore, we have now substituted this in the text for “*a thorough imprinting analysis*”.

My main concerns are as follows:

1. Only 8/25 loci are studied (with no rationale for the inclusion of only eight) of which many already have been shown to be impacted in iPSC and with current methodologies as well as the relatively low number of samples (Total N~25) the complete “catalog” should have been included.

We thank the reviewer for this valid point. In the revised version, we have now extended our analysis to 13 loci to include all the loci validated using the allele-specific version of our IMPLICON method (Klobucar et al., 2020). Unfortunately, the actual version of IMPLICON is unable to distinguish the two alleles for other imprinted loci due to lack of appropriate SNPs or inability to design robust primers. The results of the 5 new loci are very similar to the ones observed for the previous 8, strengthening our original conclusions of the effect of sex of donor cell and medium conditions on imprinting (Fig. 2B; Fig. 5A; Fig. S4E).

We would also like to point out that IMPLICON gives allele and base-pair resolution with unprecedented coverage to address methylation levels at imprinted regions. Thanks to this, we were able to uncover intra-heterogeneity of imprinting defects in some iPSC lines (Fig. S2B-C; Fig. 7B) which are novel results that were not previously discovered using other methods (bisulfite sequencing, COBRA, RRBS, bisulfite pyrosequencing, MethyIC-seq).

In the revised version, we have established 12 more iPSC lines generated using KSR/FBS and FBS+VitC protocols which makes a total of 37 samples analysed. This, by far, surpasses the number of lines used in previous studies looking at imprinting stability in mouse iPSCs (Table S1). We would also like to highlight that our F1 hybrid iPSC are isogenic and although some variability is seen, the general trends in imprinted loci were consistent taking in consideration the sex of donor cell and the reprogramming conditions. Therefore, increasing the number of iPSCs generated is unlikely to change our conclusions. The use of isogenic F1 hybrid cells enables this type of thorough analyses of reprogramming conditions otherwise

unfeasible to do with human iPSCs where a big number of lines are required to draw meaningful conclusions while controlling for their genetic variability.

2. The study is limited to mouse iPSC only despite many others have shown similar (general) phenomena in human iPSCs. The female-specific effect is not known to occur in human iPSCs.

Imprinting defects are present in human iPSCs as many papers have addressed previously (Bar et al., 2017; Ma et al., 2014; Nazor et al., 2012) and where indeed, female-specific effect is not described. Since this female-specific effect is linked to X-chromosome reactivation which only occurs in murine but not during human female iPSC reprogramming, this is unlikely to be seen in female human iPSCs. We refer this difference in the Discussion: *“For instance, human female iPSCs do not undergo X-chromosome reactivation during reprogramming (Tchieu et al., 2010) and, thus, female sex is not expected to have a major impact on imprinting defects during reprogramming of human iPSCs.”*

In our manuscript, we chose to study murine isogenic iPSCs on a hybrid background to look into imprinted regions with allele resolution thanks to the existence of SNPs. This will not be possible to be addressed in a similar comprehensive manner in human iPSCs for which we can't easily control genetic variation making mouse iPSC reprogramming the ideal system for dissecting out the contributing factors to imprinting infidelity.

3. The observed sex-specific effects in imprinting defects is not linked to differences in pluripotency or differentiation despite the observations of clear inter-individual differences in methylation aberrations within each group.

Although some inter-individual differences in methylation aberrations within each group are seen, we observe clear trends in methylation aberrations in our iPSCs as clearly depicted in Fig. 5A (female KSR- and FBS-iPSCs with general hypomethylation defects; male KSR-iPSCs with milder hypomethylation defects, male FBS-iPSCs with hypermethylation defects in *Dlk1-Dio3* locus and, to a lesser extent, in *Igf2-H19*). We have clarified in the “Results” that we see “some” and not “extensive” inter-individual differences. Importantly, the consistent trends of imprinting defects within each experimental group, is also extensive to the new KSR/FBS and FBS+VitC iPSC lines (Fig. 7A).

Our results based on the expression of pluripotent stem cell markers and teratoma assays do not show overt defects in our different iPSCs despite different types of imprinting errors (hypo- versus hypermethylation). This is mentioned in the discussion: *“that imprinting dysregulation in iPSCs does not impede the major axes of lineage commitment”*. Nonetheless, we also add now the following sentence in the discussion as a future perspective: *“Generation of chimeric mice or all-iPSC mice through blastocyst injections would provide a more refined way to address the developmental potential of distinct imprinting defects that result from the reprogramming of female and male iPSCs in different conditions and should be explored in the future in a controlled manner.”*

We now include data revealing the imprinting defects in iPSCs remain in differentiated derivatives of iPSCs (see also response to Reviewer 1, point 3). For that, we differentiated iPSCs with hypo- (F FBS5) and hypermethylation defects (M FBS5) into Neural Progenitor Cells (NPCs) and show the persistence of imprinting defects through IMPLICON and relative allele-specific expression (Fig. S8B-C). Imprinting defects as the ones we observed in iPSCs causes abnormal phenotypes in mice and imprinted syndromes in humans as we mentioned in the Discussion: *“These imprinting defects are known to cause developmental/metabolic phenotypes in mice and cause imprinted disorders in humans (Tucci et al., 2019; Monk et al., 2019)”*. Therefore, although this analysis does not directly address the impact of imprinting defects in differentiation, it shows no reversibility of imprinting defects upon differentiation which could have a negative impact in lineage decision or fitness of particular cell types as stated in the Discussion: *“Imprinting defects might rather affect specific cellular functions or differentiation of defined cellular subtypes, as it has been observed for the role of imprinted gene IGF2 in hematopoietic commitment (Nishizawa et al., 2016) or MEG3 in neural differentiation (Mo et al., 2015).”*

4. No treatment strategy or optimized culture conditions are presented rescuing the observed imprinting defects (as shown for e.g. the DLK1-Dio3 loci)

This is a very important point raised by the reviewer. For that, we concentrated on improving imprinting fidelity on male iPSCs as their defects, in contrast to female iPSCs, are linked to the reprogramming process.

Based on the knowledge about the opposite effects of KSR and FBS on imprinting in male iPSCs, we decided to test whether a 1:1 ratio of KSR/FBS or FBS with vitamin C (FBS+VitC), a component of the KSR medium that activates TET demethylase enzymes, would correct or at least mitigate the hypermethylation imprinting defects in male FBS-iPSCs. The 1:1 KSR/FBS formulation was able to fully and partially recover the hypermethylation phenotype at *Igf2-H19* and at *Dlk1-Dio3* loci, respectively. This success was offset by the hypomethylation defects on the methylated allele in the majority of loci we studied (Fig. 7A). The FBS+VitC formulation did not induce hypomethylation defects and was able to mitigate the hypermethylation defect at the *Dlk1-Dio3* locus (Fig. 7A). Close inspection of the single-molecule IMPLICON datasets showed that 3 out of 6 FBS+VitC-iPSCs have a mixed population of cells exhibiting normal and hypermethylated imprinting pattern at the *Dlk1-Dio3* locus (Fig. 7B). Using serial dilutions, we could isolate a clone presenting normal imprinting as judged by the expression of the *Meg3* gene (Fig. 7C). Curiously, this same clone also showed *H19* expression levels comparable to M FBS2 with normal imprinting at *Igf2-H19* locus (Fig. S9B). In conclusion, these data show that FBS+VitC formulation mitigates imprinting errors in male iPSCs. The concentration of VitC that we used was comparable to the ones used by Stadtfeld et al 2012 (PMID: 22387999). Their results led to a better improvement at the *Dlk1-Dio3* locus, but they did not use a sensible method as IMPLICON for this and other regions. In the discussion, we debate about these differences and suggest how playing with different VitC concentrations and other parameters could help on further improvements to produce imprinting error-free iPSCs: *“Differences in the genetic background or other technical aspects (e.g., different FBS batches) might have contributed to these differences. Playing with higher concentrations of VitC is an*

avenue to explore in the future, bearing in mind that any improvement in hypermethylation defects at the Dlk1-Dio3 or Igf2-H19 could be offset by hypomethylation defects in other imprinted loci. In conjunction to VitC adjustments, alternative reprogramming setups could also be considered such as, the use of different stoichiometry of the 4 Yamanaka factors or even the exclusion of the Oct4 from the Yamanaka cocktail which seem to have a positive impact on imprinting maintenance (Carey et al., 2011; Velychko et al., 2019)”

My additional minor concerns are as follows:

1. The authors show global changes in gene expression in female vs male iPSC:
 - a. They report that females have in general more genes upregulated than males and the authors are not reporting any discussions or interpretations why this is observed. Did they observe the same pattern for the FBS-iPSC?

Upregulation of genes in female iPSCs is due to the phenomenon of X-chromosome reactivation. Female ESCs and iPSCs are well known to have two active X chromosomes. This explains why female KSR-iPSCs and the TX1072 ESC have twice the number of X-linked reads compared to male KSR-iPSCs (Fig. S1D) and why most of the upregulated genes in female KSR-iPSCs are on the X chromosome (246 out of 413) (Fig. S1E and Table S2). It is also important to stress that clustering expression analysis of the RNA-seq data does not partition cells according to their biological sex (Fig. 1B). Therefore, the differences in the X chromosome gene dosage do not translate into major differences in global expression profile between female and male iPSCs.

The point of the reviewer concerning whether the same pattern is seen in FBS-iPSCs has now been addressed. We performed RT-qPCR analysis for three X-linked genes known to be upregulated in female KSR-iPSCs and show that the F FBS5 iPSC line has similar levels of expression compared to F KSR2 iPSC and more than male lines (M FBS1, M FBS5 and M KSR5) (Fig. S4D). Interestingly, F FBS1 iPSC shows expression levels comparable to male iPSCs, but this is because this cell line is XO, having only 39 chromosomes (Fig. S4C). Our X-painting data is also consistent with this (Fig. 1 for reviewer, below). Loss of one X chromosome is a common feature of mouse iPSCs/ESCs (Zvetkova et al., 2005 - PMID: 16244654; Pasque et al., 2018 - PMID: 29681539). Our methylation data suggest that XO karyotype follows the same female trend associated with hypomethylation defects at imprinted regions, albeit milder in certain loci (Fig. 4B).

- b. Stable 2 is incomplete as no p-values or fold-changes are reported

Due to multiple testing corrections we have not reported individual p-values for every single gene in the RNA-sequencing analyses in Table S2. Differentially expressed genes between male and female iPSCs were determined using EdgeR (p-value <0.05 with multiple testing correction) and further filtered using the intensity difference filter. This is preferred over using a fold change filter as it takes into consideration the difference in variability within the data whereby higher values show lower variability and vice versa. This information is provided in the legend of Table S2. The genes listed in the second tab of supplemental table 1, all pass these

stringent filters to be called as differentially expressed. We have then reported each individual sample values for each gene in tab3.

Reviewer #3 (Remarks to the Author):

In this manuscript, Arez et al. present a thorough analysis of imprinting integrity in mouse iPSCs by taking advantage of IMPLICON method, which gives the ability to create a base-resolution methylation map on targeted regions. The authors initially generated iPSCs from MEFs which were derived from a cross between two genetically distant mouse strains. This enabled them to discriminate between the two parental alleles when using IMPLICON and measure the allele-specific methylation levels along 8 imprinted clusters. The authors then generated iPSCs from both female and male MEFs, and under different culture conditions compared the imprinting status in each locus. They also obtained female mouse ESCs cultured under 2i, which is known to lead to global de-methylation, including in imprinted loci. These comparisons revealed a few main findings. First, the authors were able to show that when compared with their parental MEFs, iPSCs were hypomethylated in all the imprinted regions which they analyzed. They also noticed a strong bias towards a hypomethylated state in female cells, which was closer to the methylation levels observed in 2i ESCs. Next, they compared between iPSCs that were obtained and grown in KSR vs. FBS media and showed that FBS causes reduced levels of methylation-loss. This was especially evident in male iPSCs which showed methylation levels which were very close to the ones observed in their parental MEFs. Notably, the IMPLICON resolution allowed demonstrating that methylation loss was caused by heterogeneity in the culture. Importantly, the authors showed that the methylation loss in imprinted control regions resulted in the biallelic expression of imprinted genes. Finally, they showed that the differences between the sexes and media were also evident when examining global methylation levels, albeit to a lesser extent, highlighting the sensitivity of the imprinting regions for the imprint loss. The authors also looked on methylation levels along repetitive elements, which showed variable methylation states with some exhibiting similar and some distinct methylation patterns than the ones observed in the imprinted clusters.

This article uses a well-designed methodology to give a high-resolution map of imprinting aberrations in iPSCs. It also highlights the more aggressive imprinting aberrations that are caused by the KSR compared to the serum-containing (FBS) media. However, it should be noted that many of the key findings which are highlighted in this manuscript are not novel and are already well-known. Specifically, it was previously reported that female cells go through a more aggressive methylation loss in imprinted loci both in mouse ESC (Choi et al., 2017; Zvetkova et al., 2005, both referenced in the paper) and iPSC (Pasque et al., 2018, referenced in the paper). It was also reported that while male mouse ESCs grown on serum-containing media are relatively immune to imprinting aberrations, 2i ESCs (both male and female) together with female ESCs grown on serum show a global reduction of methylation that includes imprinted loci (Choi et al., 2017).

Thus, some major concerns should be addressed for this manuscript to be considered for publishing in a high-impact journal:

1. The authors point to several genes which are part of the DNA methylation machinery and are differentially expressed between iPSCs in FBS vs. KSR media, as possible drivers for the differences observed between these conditions. However, these results are only correlative. Thus, genetic perturbation or at least chemical inhibition of the suspected genes should be performed to see whether these perturbations extinguish the different methylation patterns in KSR compared with FBS. Specifically, the authors highlight the differences in the expression of Tet1/2 enzymes which were upregulated in the KSR iPSCs. They suggest that either the upregulation of Tet1/2 or their higher activity which is triggered by the presence of ascorbic acid that promotes their activity could lead to the lower methylation levels observed in the KSR media. Thus, aside for perturbing Tet1/2, the addition of ascorbic acid to the FBS media should be performed in order to see if it causes a more aggressive methylation loss.

We thank the reviewer for his important feedback. The overall goal of this study aims to understand the impact of reprogramming conditions on imprinting and find optimal protocols for the correction of these imprinting defects. As such, we were interested in playing with medium conditions rather than perturbing expression of genes fundamental for methylation/hydroxymethylation homeostasis such as the TET proteins. Indeed, there are well documented dramatic effects caused by loss of TET enzymes during reprogramming (Doege et al. Nat 2012 - PMID: 22902501; Hu et al., Cell Stem Cell 2014 - PMID: 24529596; Bartocetti et al. Cell Rep 2020 - PMID: 32187561; Caldwell et al., Mol Cell 2021 - PMID: 33352108). Moreover, KSR medium causes imprinting demethylation during maintenance of iPSCs in culture as our medium swap experiments show (Fig. S7A). Therefore, we feel it is out of scope of this manuscript to dissect the effect of *Tet1/Tet2* in the KSR medium and in the maintenance phase of iPSCs.

Based on the knowledge about the opposite effects of KSR and FBS on imprinting, we decided to test whether a mix of KSR/FBS at 1:1 ratio or FBS with ascorbic acid (FBS+VitC), as suggested by the reviewer, would correct or mitigate the hypermethylation defects in male iPSCs (female iPSCs were not tested as imprinting defects are due to culture maintenance and not reprogramming - see point 2). The 1:1 KSR/FBS formulation was able to fully and partially recover the hypermethylation phenotype at *Igf2-H19* and at *Dlk1-Dio3* loci, respectively. This success was offset by hypomethylation defects on the methylated allele in the majority of loci we studied (Fig. 7A). These hypomethylation defects were not seen for the FBS+VitC formulation, but partial hypermethylation at the *Igf2-H19* was still noticed. Importantly, FBS+VitC was able to mitigate the hypermethylation defect at the *Dlk1-Dio3* locus. Close inspection of the single-molecule IMPLICON datasets showed that 3 out of 6 FBS+VitC-iPSCs have a mixed population where some cells exhibiting normal imprinting pattern, while others exhibit a hypermethylated phenotype at the *Dlk1-Dio3* locus (Fig. 7B). Using serial dilutions, we could isolate a clone presenting normal imprinting as judged by the expression of *Meg3* gene (Fig. 7C). Curiously, this same clone also showed *H19* expression levels comparable to M FBS2 with normal imprinting at *Igf2-H19* locus (Fig. S9B). In conclusion, these data show that FBS+VitC formulation mitigates imprinting errors in male iPSCs. The concentration

of VitC that we used was comparable to the ones used by Stadtfeld et al 2012 (PMID: 22387999). Their results led to a better improvement at the *Dlk1-Dio3* locus, but they did not use a sensible method as IMPLICON for this and other regions. In the discussion, we debate about these differences and suggest how playing with different VitC concentrations and other parameters could help on further improvements that can be used to produce imprinting error-free iPSCs: “*Differences in the genetic background or other technical aspects (e.g., different FBS batches) might have contributed to these differences. Playing with higher concentrations of VitC is an avenue to explore in the future, bearing in mind that any improvement in hypermethylation defects at the Dlk1-Dio3 or Igf2-H19 could be offset by hypomethylation defects in other imprinted loci. In conjunction to VitC adjustments, alternative reprogramming setups could also be considered such as, the use of different stoichiometry of the 4 Yamanaka factors or even the exclusion of the Oct4 from the Yamanaka cocktail which seem to have a positive impact on imprinting maintenance (Carey et al., 2011; Velychko et al., 2019)*”.

2. As the sensitivity of female iPSCs and ESCs to methylation loss is already well-established, the authors should include in the analysis ESCs which are not only grown on 2i, but also on KSR and FBS. This will demonstrate whether the reprogramming of MEF to iPSCs results in higher aberration levels than what is observed in ESCs under FBS or KSR conditions. In this context, following the methylation dynamics in different time-points along the reprogramming process in every condition could strengthen the results. Following the cells along the reprogramming process will also enable to identify whether the differences in the imprinting aberrations between KSR and FBS arise at the same time-point during reprogramming or following reprogramming.

We thank the reviewer for raising this point which was also highlighted by reviewer 1. In this revised version, we have now assessed when imprinting errors occur during reprogramming. Imprinting status was analysed at day 12 and day 24 reprogramming intermediates of both male and female cells reprogrammed under the FBS protocol (Fig. 6A). We performed IMPLICON for 6 imprinted loci (Fig. 6C), one methylated and unmethylated control region (Fig. S7F; Table S3) and relative allelic expression levels using RT-PCR followed by Sanger sequencing of the *H19* and *Snrpn* imprinted genes (Fig. 6D; Fig. S7G). Moreover, we measured levels of 5mC/5hmC by LC-MS (Fig. S7E).

Concerning female iPSCs, hypomethylation imprinting errors were not apparent at day 12, with first signs of errors seen at day 24 that become clearer by day 50 (Fig. 6C). Accordingly, reactivation of the silenced allele of both *H19* and *Snrpn* is also first seen on day 24 and is only fully evident by day 50 (Fig. 6D). Therefore, hypomethylation defects in female iPSCs are uncoupled from reprogramming. Importantly, hypomethylation defects at imprinting regions did not follow the global DNA methylation dynamics during reprogramming. A decrease of 5mC is readily seen at day 12, when a peak of 5hmC and the first signs of X-chromosome reactivation are seen (Fig. 6B; Fig. S7E). To corroborate these results, the *Prickle1* methylated control region in our IMPLICON method closely follows this global DNA methylation dynamics (Fig. S7F). The fact that imprinting defects do not follow global DNA methylation dynamics and initially resist demethylation is a novel and

unexpected finding that we now highlight in the revised manuscript. Complete loss of imprinting is only seen in female “fully reprogrammed iPSCs” at day 50 (Fig. 6C), when *XIST* is no longer expressed (Fig. 6B), a hallmark of pluripotency. In conclusion, our results show that imprinted regions are initially resistant to the first demethylation wave during reprogramming, but ultimately succumb to loss of imprinting due to persistent low 5mC levels, characteristic of female murine iPSCs/ESCs (Fig. 6C; Fig. S7E) (Zvetkova et al., 2005 -PMID: 16244654; Pasque et al., 2018 -PMID: 29681539).

Concerning male cells, hypermethylation errors in the *Dlk1-Dio3* locus start to be evident at day 12 (Fig. 6C). These findings match previous results using a similar reprogramming system (Stadfeld et al, 2012). But thanks to the single-molecule resolution of IMPLICON, we gain more insights into the hypermethylation process. We observed that at day 12, in the majority of cells, the maternal ICR is hemimethylated containing both unmethylated and methylated CpGs in the same amplicon, that only at day 24 will resolve into fully methylated (and to a lesser extent fully unmethylated) state (Fig. 6E). Hypermethylation of *Dlk1-Dio3* at day 12 is also seen in female reprogramming cells as well as in both male and female non-reprogramming intermediates suggesting that it might be linked to the activation of the Yamanaka cassette. These new findings are documented in the text and depicted in Fig. 6C and reveal that hypermethylation errors at the *Dlk1-Dio3* locus are induced by reprogramming. For the *Igf2-H19* region, the mild hypermethylation errors seem to have a slightly slower kinetics, but are fully established by day 24 (Fig. 6C). Again, and in contrast to the *Prickle1* gene, these hypermethylation imprinting errors do not follow the general dynamics of DNA methylation during male reprogramming (Fig. S7E-F). These errors seem to be reprogramming-induced and explain why they have been noticed more frequently in mouse iPSCs than ESCs (Stadfeld et al., 2010 - PMID: 20418860; Yagi et al., 2019 - PMID: 31056481).

Unfortunately, we do not have the expertise in the lab to generate isogenic hybrid ESCs. Nevertheless, we wanted to address the valid point of the reviewer concerning the impact of medium conditions in imprinting stability during culture maintenance of iPSCs. For that, we decided to swap medium conditions and culture our FBS-iPSCs in KSR medium and KSR-iPSCs in FBS medium for up to 10 passages (around 20 days) and analyse by IMPLICON 6 imprinted loci, one methylated and one unmethylated control region (Fig. S7A-B; Table S3), and by relative allelic expression levels using RT-PCR followed by Sanger sequencing of the imprinted genes, *H19* and *Snrpn* (Fig. S7A-B). This experiment was actually very informative. We observed that culturing FBS-iPSCs in KSR medium resulted in imprinting instability that ranges from milder to stronger effects (Fig. S7A). This was also the reason why we did not investigate imprinting defects during KSR derivation of iPSCs, since the KSR medium by itself induces imprinting errors. We observed the opposite when we swapped the medium to FBS, as we did not see differences in methylation or relative allelic expression of imprinted genes in the time frame of this experiment (Fig. S7B).

3. The authors also point to X-linked genes which were differentially expressed between female and male iPSCs as possible drivers for the methylation difference between them. However, these results are also correlative and should be established by perturbation of the genes.

Our results suggest the hypomethylation phenotype in female cells is uncoupled from reprogramming and occurs during prolonged culture of these cells and presumably a passive consequence of X-chromosome reactivation and increased dosage of X-linked genes (Fig. S1D-E; Fig. S4D; Fig. 6B). Candidate X-linked genes involved in the global hypomethylation phenotype have been previously identified (Choi et al., 2017. PMID: 28366588; Pasque et al., 2018 - PMID: 29681539). Those genes are likely to be responsible for hypomethylation at imprinted genes, but it is out of scope of our study to dissect the molecular players involved as these errors occurred after reprogramming. Our feeling is that the only way to escape the hypomethylation defects in female stem cells would be to generate induced epiblast stem cells (EpiSCs), which, in theory, would not undergo X-chromosome reactivation. We allude to this possibility in the Discussion section: *“Higher dosage of X-linked genes due to the presence of two active X-chromosomes have been implicated in this phenotype. In particular, the X-linked Dusp9 gene has been implicated in hypomethylation of female stem cells (Choi et al., 2017a), however, other X-linked genes might also be involved during reprogramming (Song et al., 2019). As X-chromosome reactivation is a hallmark of female iPSCs, strategies to protect imprints by manipulating gene expression from X-linked genes need to be envisioned. Alternatively, as epiblast stem cells (EpiSCs) retain an inactive X-chromosome, the generation of induced EpiSCs (iEpiSCs) (Han et al. 2012) might provide an alternative to produce female pluripotent stem cells devoid of imprinting defects. A thorough analysis on X-chromosome status during iEpiSC derivation will provide insights about the feasibility of this strategy.”*

4. In Figure S3A, the percentage of reads obtained from each chromosome in each cell-line is presented. However, this presentation does not give the ability to fully assess whether the cells have a normal karyotype. In this context, it is not clear why the male iPSCs do not have any reads obtained from the Y chromosome. Thus, a karyotype for each line should be provided. This will also enable to understand whether imprinting aberrations are connected to chromosomal instability.

We thank the reviewer for raising this very important point. We have now provided karyotype for the 8 main KSR- and FBS-iPSCs (Fig. S1C and Fig. S4C) In general, our iPSCs show a normal karyotype. An exception to that is the F FBS1 iPSC line which showed to be XO, based on a karyotype of 39 chromosomes (Fig. S4C), X-linked gene expression level comparable to male iPSCs (Fig. S4D) and X-painting data showing the presence of only one X on a metaphase spread (Fig. 1 for reviewers). Loss of one X chromosome is a common feature of mouse iPSCs/ESCs (Zvetkova et al., 2005 -PMID: 16244654; Pasque et al., 2018 -PMID: 29681539). Our methylation data suggest that XO karyotype is still associated with hypomethylation defects at imprinted regions, with milder effects seen in a few imprinted regions (Fig. 4B; Fig. 5A; Table S3). Unfortunately, we cannot trace back when the X chromosome was lost in this line. Concerning the F KSR 2 line which exhibits loss of paternal chromosome 6 (Fig. S3A), we also detected the presence of a normal karyotype (Fig. S1C). Based on allele-specific read counts on our RNAseq analysis, we showed that absence of paternal chromosome 6 was compensated by the presence of two maternal chromosomes 6. This data is provided in Fig. S3B. Overall, our iPSCs show

a normal karyotype which suggests that chromosomal instability is not the underlying cause of imprinting aberrations seen in our iPSCs. This is consistent with the fact that imprinting errors were dictated by the sex of donor cell and reprogramming conditions with minimal heterogeneity between independent iPSC lines.

Concerning the read counts of the Y chromosome in Fig. S3A, we only represent reads from high-confidence SNPs from the mouse genome. The Y chromosome has a rather low number of these positions (only 1132 SNPs; the X-chromosome has 636442 SNPs for example). This explains the low number attributed to the Y chromosome in male iPSCs. We added a sentence to the legend of this figure to clarify this point.

5. In Figure S5B, the expression of *Meg3* for every iPSC line is shown in order to establish that lines which showed more aggressive methylation erasure in their imprinting control regions also express higher levels of imprinted genes. However, this should be performed for other genes and not only for *Meg3*. Furthermore, the correlation between the methylation in an imprinting control region to the expression levels of genes under the control of the region should be plotted as a correlation plot which is much easier to interpret.

We thank the reviewer for raising this point. We have now made a correlation plot of expression vs methylation for the *Meg3* as suggested by the reviewer to show nicely a negative correlation between these two variables (Fig. S5B). We also made correlation plots for *H19* and *Peg3* for FBS-iPSCs and KSR-iPSCs based on RT-qPCR data as well as for *Zrsr1* and *Snrpn* genes for the male and female KSR-iPSCs based on RNAseq data. This is provided as Fig. 2 for reviewers. For all these genes, a negative correlation was also observed, although less visible than for the *Meg3* gene (Fig. S5B). This can be explained by the fact that methylation oscillations for these loci have a tighter dynamic range than for *Dlk1-Dio3* locus, where *Meg3* is located. We decided not to include these data in the paper due to size constraints.

6. In this manuscript the focus was on 8 imprinted clusters while the original IMPLICON paper included 9 clusters (Klobučar et al., 2020, referenced in the manuscript). The authors should therefore explain why they did not include the last region that was included in the first paper.

This is a valid point from the reviewer that was also mentioned by reviewer 2. As a matter of fact, the allele-specific IMPLICON method covered 13 imprinted regions (Klobučar et al., 2020 - PMID: 32621604). To satisfy this important point, we had now added 5 extra loci (*Mest/Peg1*, *Plagl1/Zac1*, *Grb10*, *Igf2r* and *Impact*) which were missing in our original dataset. The results of the 5 new loci are very similar to the ones observed for the previous 8 (Fig. 2B; Fig. S4E; Fig. 5A), strengthening our original conclusions of the effect of sex of donor cell and medium conditions on imprinting fidelity.

Minor comment:

1. Figures S6A, S6B should have separate legends.

I guess the reviewer meant Fig. 6A and 6C (they are now Fig. 5A and 5C). This is now fixed in the revised manuscript.

Figure 1 for Reviewers - Representative X-chromosome painting images of F FBS1 (XO) and F FBS5 iPSCs (XX).

Mitotic chromosomes are marked in blue by DAPI and X chromosomes are marked in green (one for F FBS1 and two for F FBS5); Scale bars correspond to 11 µm.

Figure 2 for Reviewers - Correlation between DNA methylation at ICRs and expression of imprinted genes.

- A. Scatter plots representing the correlation between ICR methylation at the *Igf2-H19* and *Peg3* loci and imprinted gene expression of, respectively, *H19* and *Peg3* genes for the samples: F KSR2, F KSR4, M KSR3, M KSR5, F FBS1, F FBS5, M FBS1 and M FBS5. The X-axis represents the average methylation levels of ICRs considering both parental alleles. The Y-axis represents the *H19/Gapdh* or *Peg3/Gapdh* values measured by RT-qPCR expression analysis and normalised to the M FBS1 iPSC line ($n=3$). r represents the Pearson's correlation between methylation levels of the respective ICRs measured by IMPLICON and expression levels measured by RT-qPCR.
- B. Scatter plots representing the correlation between ICR methylation at the *Commd1-Zrsr1* and PWS/AS loci and imprinted gene expression of, respectively, *Zrsr1* and *Snrpn* genes for the samples: F KSR2, F KSR4, M KSR3 and M KSR5. The X-axis represents the average methylation levels of the ICRs considering both parental alleles. The Y-axis represents the average Log2 Reads per kilobase per million mapped reads (RPKM) expression values for the *Zrsr1* and *Snrpn* genes from biological triplicates of each sample measured by RNAseq. r represents the Pearson's correlation between methylation levels of the respective ICRs measured by IMPLICON and expression levels measured by RNAseq.

REVIEWERS' COMMENTS

Reviewer #1 (Remarks to the Author):

I have carefully evaluated the revised manuscript by Arez and colleague (now called "Imprinting fidelity in mouse iPSCs depends on sex of donor cell and medium formulation") and would like to congratulate the authors on doing an excellent job at addressing my concerns. While the authors opted not to conduct all experiments that I had suggested (i.e. blastocyst injections) their reasons for doing so are valid and the alternative experiments they have included instead are informative. I have no further concerns.

Reviewer #2 (Remarks to the Author):

I have no additional request.

Reviewer #3 (Remarks to the Author):

The authors have addressed my comments.